# Convergence of Mean-field Langevin dynamics: Time-space discretization, stochastic gradient, and variance reduction

**Taiji Suzuki**[1,2], **Denny Wu**[3,4], **Atsushi Nitanda**[2,5]

[1]University of Tokyo,  [2]RIKEN AIP,  [3]New York University,
[4]Flatiron Institute,  [5]Kyushu Institute of Technology

taiji@mist.i.u-tokyo.ac.jp, dennywu@nyu.edu, nitanda@ai.kyutech.ac.jp

## Abstract

The mean-field Langevin dynamics (MFLD) is a nonlinear generalization of the Langevin dynamics that incorporates a distribution-dependent drift, and it naturally arises from the optimization of two-layer neural networks via (noisy) gradient descent. Recent works have shown that MFLD globally minimizes an entropy-regularized convex functional in the space of measures. However, all prior analyses assumed the infinite-particle or continuous-time limit, and cannot handle stochastic gradient updates. We provide a general framework to prove a uniform-in-time propagation of chaos for MFLD that takes into account the errors due to finite-particle approximation, time-discretization, and stochastic gradient. To demonstrate the wide applicability of our framework, we establish quantitative convergence rate guarantees to the regularized global optimal solution for $(i)$ a wide range of learning problems such as mean-field neural network and MMD minimization, and $(ii)$ different gradient estimators including SGD and SVRG. Despite the generality of our results, we achieve an improved convergence rate in both the SGD and SVRG settings when specialized to the standard Langevin dynamics.

## 1 Introduction

In this work we consider the mean-field Langevin dynamics (MFLD) given by the following McKean-Vlasov stochastic differential equation:

$$\mathrm{d}X_t = -\nabla\frac{\delta F(\mu_t)}{\delta\mu}(X_t)\mathrm{d}t + \sqrt{2\lambda}\mathrm{d}W_t, \tag{1}$$

where $\mu_t = \mathrm{Law}(X_t)$, $F : \mathcal{P}_2(\mathbb{R}^d) \to \mathbb{R}$ is a convex functional, $W_t$ is the $d$-dimensional standard Brownian motion, and $\frac{\delta F}{\delta\mu}$ denotes the first-variation of $F$. Importantly, MFLD is the Wasserstein gradient flow that minimizes an entropy-regularized convex functional as follows:

$$\min_{\mu\in\mathcal{P}_2}\{F(\mu) + \lambda\mathrm{Ent}(\mu)\}. \tag{2}$$

While the above objective can also be solved using other methods such as double-loop algorithms based on iterative linearization (Nitanda et al., 2020, 2023; Oko et al., 2022), MFLD remains attractive due to its simple structure and connection to neural network optimization. Specifically, the learning of two-layer neural networks can be lifted into an infinite-dimensional optimization problem in the space of measures (i.e., the *mean-field limit*), for which the convexity of loss function can be exploited to show the global convergence of gradient-based optimization (Nitanda and Suzuki, 2017; Mei et al., 2018; Chizat and Bach, 2018; Rotskoff and Vanden-Eijnden, 2018; Sirignano and Spiliopoulos, 2020). Under this viewpoint, MFLD Eq. (1) corresponds to the continuous-time limit of the *noisy* gradient descent update on an infinite-width neural network, where the injected Gaussian noise encourages "exploration" and facilities global convergence (Mei et al., 2018; Hu et al., 2019).

37th Conference on Neural Information Processing Systems (NeurIPS 2023).

**Quantitative analysis of MFLD.**  Most existing convergence results of neural networks in the mean-field regime are *qualitative* in nature, that is, they do not characterize the rate of convergence nor the discretization error. A noticeable exception is the recent analysis of MFLD by Nitanda et al. (2022); Chizat (2022), where the authors proved exponential convergence to the optimal solution of Eq. (2) under a *logarithmic Sobolev inequality* (LSI) that can be verified in various settings including regularized empirical risk minimization using neural networks. This being said, there is still a large gap between the ideal MFLD analyzed in prior works and a feasible algorithm. In practice, we parameterize $\mu$ as a mixture of $N$ particles $(X^i)_{i=1}^N$ — this corresponds to a neural network with $N$ neurons, and perform a discrete-time update: at time step $k$, the update to the $i$-th particle is given as

$$X_{k+1}^i = X_k^i - \eta_k \tilde{\nabla} \frac{\delta F(\mu_k)}{\delta \mu}(X_k^i) + \sqrt{2\lambda\eta_k}\xi_k^i, \tag{3}$$

where $\eta_k$ is the step size at the $k$-th iteration, $\xi_k^i$ is an i.i.d. standard Gaussian vector, and $\tilde{\nabla}\frac{\delta F(\mu)}{\delta \mu}$ represents a potentially inexact (e.g., stochastic) gradient.

Comparing Eq. (1) and Eq. (3), we observe the following discrepancies between the ideal MFLD and the implementable noisy particle gradient descent algorithm.

(i) **Particle approximation.** $\mu$ is entirely represented by a finite set of particles: $\mu_k = \frac{1}{N}\sum_{i=1}^N \delta_{X_k^i}$.

(ii) **Time discretization.** We employ discrete gradient descent update as opposed to gradient flow.

(iii) **Stochastic gradient.** In many practical settings, it is computationally prohibitive to obtain the exact gradient update, and hence it is preferable to adopt a stochastic estimate of the gradient.

The control of finite-particle error (point $(i)$) is referred to as *propagation of chaos* (Sznitman, 1991) (see also Lacker (2021) and references therein). In the context of mean-field neural networks, discretization error bounds in prior works usually grow *exponentially in time* (Mei et al., 2018; Javanmard et al., 2019; De Bortoli et al., 2020), unless one introduces additional assumptions on the dynamics that are difficult to verify (Chen et al., 2020). Consequently, convergence guarantee in the continuous limit cannot be transferred to the finite-particle setting unless the time horizon is very short (e.g., Abbe et al. (2022)), which limits the applicability of the theory.

Very recently, Chen et al. (2022); Suzuki et al. (2023) established a *uniform-in-time* propagation of chaos for MFLD, i.e., the "distance" between the $N$-particle system and the infinite-particle limit is of order $\mathcal{O}(1/N)$ for all $t > 0$. While this represents a significant step towards an optimization theory for practical finite-width neural networks in the mean-field regime, these results assumed the continuous-time limit and access to exact gradient, thus cannot cover points $(ii)$ and $(iii)$.

In contrast, for the standard gradient Langevin dynamics (LD) without the mean-field interactions (which is a special case of MFLD Eq. (1) by setting $F$ to be a *linear* functional), the time discretization is well-understood (see e.g. Dalalyan (2014); Vempala and Wibisono (2019); Chewi et al. (2021)), and its stochastic gradient variant (Welling and Teh, 2011; Ma et al., 2015), including ones that employ variance-reduced gradient estimators (Dubey et al., 2016; Zou et al., 2018; Kinoshita and Suzuki, 2022), have also been extensively studied.

The gap between convergence analyses of LD and MFLD motivates us to ask the following question.

*Can we develop a complete non-asymptotic convergence theory for MFLD that takes into account points $(i)$ - $(iii)$, and provide further refinement over existing results when specialized to LD?*

## 1.1 Our Contributions

We present a unifying framework to establish uniform-in-time convergence guarantees for the mean-field Langevin dynamics under time and space discretization, simultaneously addressing points $(i)$-$(iii)$. The convergence rate is exponential up to an error that vanishes when the step size and stochastic gradient variance tend to $0$, and the number of particles $N$ tends to infinity. Moreover, our proof is based on an LSI condition analogous to Nitanda et al. (2022); Chizat (2022), which is satisfied in a wide range of regularized risk minimization problems. The advantages of our analysis is summarized as follows.

- Our framework provides a unified treatment of different gradient estimators. Concretely, we establish convergence rate of MFLD with stochastic gradient and stochastic variance reduced gradient (Johnson and Zhang, 2013). While it is far from trivial to derive a tight bound on the stochastic

| Method (authors) | # of particles | Total complexity | Single loop | Mean-field |
|---|---|---|---|---|
| PDA* (Nitanda et al., 2021) | $\epsilon^{-2}\log(n)$ | $G_\epsilon \epsilon^{-1}$ | × | ✓ |
| P-SDCA (Oko et al., 2022) | $\epsilon^{-1}\log(n)$ | $G_\epsilon(n+\frac{1}{\lambda})\log(\frac{n}{\epsilon})$ | × | ✓ |
| GLD (Vempala and Wibisono, 2019) | — | $\frac{n}{\epsilon}\frac{\log(\epsilon^{-1})}{(\lambda\alpha)^2}$ | ✓ | × |
| SVRG-LD (Kinoshita and Suzuki, 2022) | — | $\left(n+\frac{\sqrt{n}}{\epsilon}\right)\frac{\log(\epsilon^{-1})}{(\lambda\alpha)^2}$ | ✓ | × |
| F-MFLD (ours) | $\epsilon^{-1}$ | $nE_*\frac{\log(\epsilon^{-1})}{(\lambda\alpha)}$ | ✓ | ✓ |
| SGD-MFLD* (ours) | $\epsilon^{-1}$ | $\epsilon^{-1}E_*\frac{\log(\epsilon^{-1})}{(\lambda\alpha)}$ | ✓ | ✓ |
| SGD-MFLD* (ii) (ours) | $\epsilon^{-1}$ | $\epsilon^{-1}(1+\sqrt{\lambda E_*})\frac{\log(\epsilon^{-1})}{(\lambda\alpha)^2}$ | ✓ | ✓ |
| SVRG-MFLD (ours) | $\epsilon^{-1}$ | $\sqrt{n}E_*\frac{\log(\epsilon^{-1})}{(\lambda\alpha)}+n$ | ✓ | ✓ |
| SVRG-MFLD (ii) (ours) | $\epsilon^{-1}$ | $(n^{1/3}E_*+\sqrt{n}\lambda^{1/4}E_*^{3/4})\frac{\log(\epsilon^{-1})}{(\lambda\alpha)}+n$ | ✓ | ✓ |

Table 1: Comparison of computational complexity to optimize an entropy-regularized finite-sum objective up to excess objective value $\epsilon$, in terms of dataset size $n$, entropy regularization $\lambda$, and LSI constant $\alpha$. Label * indicates the *online* setting, and the unlabeled methods are tailored to the *finite-sum* setting. "Mean-field" indicates the presence of particle interactions. "Single loop" indicates whether the algorithm requires an inner-loop MCMC sampling sub-routine at every step. "(ii)" indicates convergence rate under additional smoothness condition (Assumption 4), where $E_* = \frac{\bar{L}^2}{\alpha\epsilon} + \frac{\bar{L}}{\sqrt{\lambda\alpha\epsilon}}$. For double-loop algorithms (PDA and P-SDCA), $G^*$ is the number of gradient evaluations required for MCMC sampling; for example, for MALA (Metropolis-adjusted Langevin algorithm) $G_\epsilon = O(n\alpha^{-5/2}\log(1/\epsilon)^{3/2})$, and for LMC (Langevin Monte Carlo) $G_\epsilon = O(n(\alpha\epsilon)^{-2}\log(\epsilon))$.

gradient approximation error because it requires evaluating the correlation between the randomness of each gradient and the updated parameter distribution, we are able to show that a stochastic gradient effectively improves the computational complexity.

Noticeably, despite the fact that our theorem simultaneously handles a wider class of $F$, when specialized to standard Langevin dynamics (i.e., when $F$ is linear), we recover state-of-the-art convergence rates for LD; moreover, by introducing an additional mild assumption on the smoothness of the objective, our analysis can significantly improve upon existing convergence guarantees.

- Our analysis greatly extends the recent works of Chen et al. (2022); Suzuki et al. (2023), in that our propagation of chaos result covers the discrete time setting while the discretization error can still be controlled uniformly over time, i.e., the finite particle approximation error does not blow up as $t$ increases. Noticeably, we do not impose weak interaction / large noise conditions that are common in the literature (e.g., Delarue and Tse (2021); Lacker and Flem (2022)); instead, our theorem remains valid for *any* regularization strength.

## 2  Problem Setting

Consider the set of probability measure $\mathcal{P}$ on $\mathbb{R}^d$ where the Borel $\sigma$-algebra is equipped. Our goal is to find a probability measure $\mu \in \mathcal{P}$ that approximately minimizes the objective given by

$$F(\mu) = U(\mu) + \mathbb{E}_\mu[r],$$

where $U : \mathcal{P} \to \mathbb{R}$ is a (convex) loss function, and $r : \mathbb{R}^d \to \mathbb{R}$ is a regularization term. Let $\mathcal{P}_2$ be the set of probability measures with the finite second moment. In the following, we consider the setting where $F(\mu) \le C(1 + \mathbb{E}_\mu[\|X\|^2])$, and focus on $\mathcal{P}_2$ so that $F$ is well-defined.

As previously mentioned, an important application of this minimization problem is the learning of two-layer neural network in the *mean-field regime*. Suppose that $h_x(\cdot)$ is a neuron with a parameter $x \in \mathbb{R}^d$, e.g., $h_x(z) = \sigma(w^\top z + b)$, where $w \in \mathbb{R}^{d-1}, b \in \mathbb{R}$, and $x = (w, v)$. The mean-field neural network corresponding to a probability measure $\mu \in \mathcal{P}$ can be written as $f_\mu(\cdot) = \int h_x(\cdot)\mu(\mathrm{d}x)$. Given training data $(z_i, y_i)_{i=1}^n \in \mathbb{R}^{d-1} \times \mathbb{R}$, we may define the empirical risk of $f_\mu$ as $U(\mu) = \frac{1}{n}\sum_{i=1}^n \ell(f_\mu(z_i), y_i)$ for a loss function $\ell : \mathbb{R} \times \mathbb{R} \to \mathbb{R}$. Then, the objective $F$ becomes

$$F(\mu) = \frac{1}{n}\sum_{i=1}^n \ell(f_\mu(z_i), y_i) + \lambda_1 \int \|x\|^2 \mathrm{d}\mu(x),$$

where the regularization term is $r(x) = \lambda_1 \|x\|^2$. Note that the same objective can be defined for expected risk minimization. We defer additional examples to the last of this section.

One effective way to solve the above objective is the *mean-field Langevin dynamics* (MFLD), which optimizes $F$ via a noisy gradient descent update. To define MFLD, we need to introduce the first-variation of the functional $F$.

**Definition 1.** *Let $G : \mathcal{P}_2 \to \mathbb{R}$. The first-variation $\frac{\delta G}{\delta \mu}$ of a functional $G : \mathcal{P}_2 \to \mathbb{R}$ at $\mu \in \mathcal{P}_2$ is defined as a continuous functional $\mathcal{P}_2 \times \mathbb{R}^d \to \mathbb{R}$ that satisfies $\lim_{\epsilon \to 0} \frac{G(\epsilon\nu + (1-\epsilon)\mu)}{\epsilon} = \int \frac{\delta G}{\delta \mu}(\mu)(x)\mathrm{d}(\nu - \mu)$ for any $\nu \in \mathcal{P}_2$. If there exists such a functional $\frac{\delta G}{\delta \mu}$, we say $G$ admits a first-variation at $\mu$, or simply $G$ is differentiable at $\mu$.*

To avoid the ambiguity of $\frac{\delta G}{\delta \mu}$ up to constant shift, we follow the convention of imposing $\int \frac{\delta G}{\delta \mu}(\mu)\mathrm{d}\mu = 0$. Using the first-variation of $F$, the MFLD is given by the following stochastic differential equation:

$$\mathrm{d}X_t = -\nabla \frac{\delta F(\mu_t)}{\delta \mu}(X_t)\mathrm{d}t + \sqrt{2\lambda}\mathrm{d}W_t, \quad \mu_t = \mathrm{Law}(X_t), \tag{4}$$

where $X_0 \sim \mu_0$, $\mathrm{Law}(X)$ denotes the distribution of the random variable $X$ and $(W_t)_{t\geq 0}$ is the $d$-dimensional standard Brownian motion. Readers may refer to Huang et al. (2021) for the existence and uniqueness of the solution. MFLD is an instance of *distribution-dependent* SDE because the drift term $\frac{\delta F(\mu_t)}{\delta \mu}(\cdot)$ depends on the distribution $\mu_t$ of the current solution $X_t$ (Kahn and Harris, 1951; Kac, 1956; McKean, 1966). It is known that the MFLD is a Wasserstein gradient flow to minimize the following *entropy-regularized* objective (Mei et al., 2018; Hu et al., 2019; Nitanda et al., 2022; Chizat, 2022):

$$\mathscr{F}(\mu) = F(\mu) + \lambda\mathrm{Ent}(\mu), \tag{5}$$

where $\mathrm{Ent}(\mu) = -\int \log(\mathrm{d}\mu(z)/\mathrm{d}z)\mathrm{d}\mu(z)$ is the negative entropy of $\mu$.

**Reduction to standard Langevin dynamics.** Note that the MFLD reduces to the standard gradient Langevin dynamics (LD) when $F$ is a linear functional, that is, there exists $V$ such that $F(\mu) = \int V(x)\mathrm{d}\mu(x)$. In this case, $\frac{\delta F}{\delta \mu} = V$ for any $\mu$ and the MFLD Eq. (4) simplifies to

$$\mathrm{d}X_t = -\nabla V(X_t)\mathrm{d}t + \sqrt{2\lambda}\mathrm{d}W_t.$$

This is exactly the gradient Langevin dynamics for optimizing $V$ or sampling from $\mu \propto \exp(-V/\lambda)$.

## 2.1 Some Applications of MFLD

Here, we introduce a few examples that can be approximately solved via MFLD.

*Example* 1 (Two-layer neural network in mean-field regime.). Let $h_x(z)$ be a neuron with a parameter $x \in \mathbb{R}^d$, e.g., $h_x(z) = \tanh(r\sigma(w^\top x))$, $h_x(z) = \tanh(r)\sigma(w^\top x)$ for $x = (r, w)$, or simply $h_x(z) = \sigma(x^\top z)$. Then the learning of mean-field neural network $f_\mu(\cdot) = \int h_x(\cdot)\mu(\mathrm{d}x)$ via minimizing the empirical risk $U(\mu) = \frac{1}{n}\sum_{i=1}^n \ell(f_\mu(z_i), y_i)$ with a convex loss (e.g., the logistic loss $\ell(f, y) = \log(1 + \exp(-yf))$, or the squared loss $\ell(f, y) = (f - y)^2$) falls into our framework.

*Example* 2 (Density estimation via MMD minimization). For a positive definite kernel $k$, the Maximum Mean Discrepancy (MMD) (Gretton et al., 2006) between two probability measures $\mu$ and $\nu$ is defined as $\mathrm{MMD}^2(\mu, \nu) := \int\int(k(x, x) - 2k(x, y) + k(y, y))\mathrm{d}\mu(x)\mathrm{d}\nu(y)$. We perform nonparametric density estimation by fitting a Gaussian mixture model $f_\mu(z) = \int g_x(z)\mathrm{d}\mu(x)$, where $g_x$ is the Gaussian density with mean $x$ and a given variance $\sigma^2 > 0$. The mixture model is learned by minimizing $\mathrm{MMD}^2(f_\mu, p^*)$ where $p^*$ is the target distribution. If we observe a set of training data $(z_i)_{i=1}^n$ from $p^*$, then the empirical version of MMD is one suitable loss function $U(\mu)$ given as

$$\widehat{\mathrm{MMD}}^2(\mu) := \int\left[\int\int g_x(z)g_{x'}(z')k(z, z')\mathrm{d}z\mathrm{d}z'\right]\mathrm{d}(\mu \times \mu)(x, x') - 2\int\left(\frac{1}{n}\sum_{i=1}^n \int g_x(z)k(z, z_i)\mathrm{d}z\right)\mathrm{d}\mu(x),$$

Note that we may also choose to directly fit the particles to the data (instead of a Gaussian mixture model), that is, we use a Dirac measure for each particle, as in Chizat (2022, Section 5.2) and Arbel et al. (2019). Here we state the Gaussian parameterization for the purpose of density estimation.

*Example* 3 (Kernel Stein discrepancy minimization). In settings where we have access to the target distribution through the score function (e.g., sampling from a posterior distribution in Bayesian inference), we may employ the kernel Stein discrepancy (KSD) as the discrepancy measure (Chwialkowski et al., 2016; Liu et al., 2016). Suppose that we can compute $s(z) = \nabla \log(\mu^*)(z)$, then for a positive definite kernel $k$, we define the *Stein kernel* as

$$W_{\mu^*}(z, z') := s(z)^\top s(z')k(z, z') + s(z)^\top \nabla_{z'} k(z, z') + \nabla_z^\top k(z, z')s(z') + \nabla_{z'}^\top \nabla_z k(z, z').$$

We take $U(\mu)$ as the KSD between $\mu$ and $\mu^*$ defined as $\mathrm{KSD}(\mu) = \int \int W_{\mu^*}(z, z')\mathrm{d}\mu(z)\mathrm{d}\mu(z')$. By minimizing this objective via MFLD, we attain kernel Stein variational inference with convergence guarantees. This can be seen as a "Langevin" version of KSD descent in Korba et al. (2021).

**Remark 1.** *In the above examples, we introduce additional regularization terms in the objective Eq. (5) to establish the convergence rate of MFLD, in exchange for a slight optimization bias. Note that these added regularizations often have a statistical benefit due to the smoothing effect.*

## 2.2 Practical implementations of MFLD

Although the convergence of MFLD (Eq. (4)) has been studied in prior works (Hu et al., 2019; Nitanda et al., 2022; Chizat, 2022), there is a large gap between the ideal dynamics and a practical implementable algorithm. Specifically, we need to consider $(i)$ the finite-particle approximation, $(ii)$ the time discretization, and $(iii)$ stochastic gradient.

To this end, we consider the following space- and time-discretized version of the MFLD with stochastic gradient update. For a finite set of particles $\mathscr{X} = (X^i)_{i=1}^N \subset \mathbb{R}^d$, we define its corresponding empirical distribution as $\mu_{\mathscr{X}} = \frac{1}{N}\sum_{i=1}^N \delta_{X_i}$. Let $\mathscr{X}_k = (X_k^i)_{i=1}^N \subset \mathbb{R}^d$ be $N$ particles at the $k$-th update, and define $\mu_k = \mu_{\mathscr{X}_k}$ as a finite particle approximation of the population counterpart. Starting from $X_0^i \sim \mu_0$, we update $\mathscr{X}_k$ as,

$$X_{k+1}^i = X_k^i - \eta_k v_k^i + \sqrt{2\lambda\eta_k}\xi_k^i, \tag{6}$$

where $\eta_k > 0$ is the step size, $\xi_k^i$ is an i.i.d. standard normal random variable $\xi_k^i \sim N(0, I)$, and $v_k^i = v_k^i(\mathscr{X}_{0:k}, \omega_k^i)$ is a stochastic approximation of $\nabla \frac{\delta F(\mu_k)}{\delta \mu}(X_k^i)$ where $\omega_k = (\omega_k^i)_{i=1}^N$ is a random variable generating the randomness of stochastic gradient. $v_k^i$ can depend on the history $\mathscr{X}_{0:k} = (\mathscr{X}_0, \dots, \mathscr{X}_k)$, and $\mathbb{E}_{\omega_k^i}[v_k^i | \mathscr{X}_{0:k}] = \nabla \frac{\delta F(\mu_k)}{\delta \mu}(X_k^i)$. We analyze three versions of $v_k^i$.

**(1) Full gradient: F-MFLD.** If we have access to the exact gradient, we may compute

$$v_k^i = \nabla \frac{\delta F(\mu_k)}{\delta \mu}(X_k^i).$$

**(2) Stochastic gradient: SGD-MFLD.** Suppose the loss function $U$ is given by an expectation as $U(\mu) = \mathbb{E}_{Z \sim P_Z}[\ell(\mu, Z)]$, where $Z \in \mathcal{Z}$ is a random observation obeying a distribution $P_Z$ and $\ell : \mathcal{P} \times \mathcal{Z} \to \mathbb{R}$ is a loss function. In this setting, we construct the stochastic gradient as

$$v_k^i = \frac{1}{B}\sum_{j=1}^B \nabla \frac{\delta \ell(\mu_k, z_k^{(j)})}{\delta \mu}(X_k^i) + \nabla r(X_k^i),$$

where $(z_k^{(j)})_{j=1}^B$ is a mini-batch of size $B$ generated from $P_Z$ in an i.i.d. manner.

**(3) Stochastic variance reduced gradient: SVRG-MFLD.** Suppose that the loss function $U$ is given by a finite sum of loss functions $\ell_i$: $U(\mu) = \frac{1}{n}\sum_{i=1}^n \ell_i(\mu)$, which corresponds to the empirical risk in a usual machine learning setting. Then, the variance reduced stochastic gradient (SVRG) (Johnson and Zhang, 2013) is defined as follows:

(i) When $k \equiv 0 \mod m$, where $m$ is the update frequency, we set $\dot{\mathscr{X}} = (\dot{X}^i)_{i=1}^n = (X_k^i)_{i=1}^n$ as the anchor point and refresh the stochastic gradient as $v_k^i = \nabla \frac{\delta F(\mu_k)}{\delta \mu}(X_k^i)$.

(ii) When $k \not\equiv 0 \mod m$, we use the anchor point to construct a control variate and compute

$$v_k^i = \frac{1}{B}\sum_{j \in I_k} \nabla \left( \frac{\delta \ell_j(\mu_{\mathscr{X}_k})}{\delta \mu}(X_k^i) + r(X_k^i) - \frac{\delta \ell_j(\mu_{\dot{\mathscr{X}}})}{\delta \mu}(\dot{X}^i) + \frac{\delta U(\mu_{\dot{\mathscr{X}}})}{\delta \mu}(\dot{X}^i) \right),$$

where $I_k$ is a uniformly drawn random subset of $\{1, \dots, n\}$ with size $B$ without duplication.

# 3 Main Assumptions and Theoretical Tools

For our convergence analysis, we make the following assumptions which are inherited from prior works in the literature (Nitanda et al., 2020, 2022; Chizat, 2022; Oko et al., 2022; Chen et al., 2022).

**Assumption 1.** *The loss function $U$ and the regularization term $r$ are convex. Specifically,*

1. *$U : \mathcal{P} \to \mathbb{R}$ is a convex functional on $\mathcal{P}$, that is, $U(\theta\mu + (1-\theta)\nu) \leq \theta U(\mu) + (1-\theta)U(\nu)$ for any $\theta \in [0,1]$ and $\mu, \nu \in \mathcal{P}$. Moreover, $U$ admits a first-variation at any $\mu \in \mathcal{P}_2$.*

2. *$r(\cdot)$ is twice differentiable and convex, and there exist constants $\lambda_1, \lambda_2 > 0$ and $c_r > 0$ such that $\lambda_1 I \preceq \nabla\nabla^\top r(x) \preceq \lambda_2 I$, $x^\top \nabla r(x) \geq \lambda_1\|x\|^2$, and $0 \leq r(x) \leq \lambda_2(c_r + \|x\|^2)$ for any $x \in \mathbb{R}^d$, and $\nabla r(0) = 0$.*

**Assumption 2.** *There exists $L > 0$ such that $\left\| \nabla \frac{\delta U(\mu)}{\delta\mu}(x) - \nabla \frac{\delta U(\mu')}{\delta\mu}(x') \right\| \leq L(W_2(\mu, \mu') + \|x - x'\|)$ and $\left| \frac{\delta^2 U(\mu)}{\delta\mu^2}(x, x') \right| \leq L(1 + c_L(\|x\|^2 + \|x'\|^2))$ for any $\mu, \mu' \in \mathcal{P}_2$ and $x, x' \in \mathbb{R}^d$. Also, there exists $R > 0$ such that $\|\nabla\frac{\delta U(\mu)}{\delta\mu}(x)\| \leq R$ for any $\mu \in \mathcal{P}$ and $x \in \mathbb{R}^d$.*

Verification of this assumption in the three examples (Examples 1, 2 and 3) is given in Appendix A in the supplementary material. We remark that the assumption on the second order variation is only required to derive the discretization error corresponding to the uniform-in-time propagation of chaos (Lemma 8 in Appendix E.4). Under these assumptions, Proposition 2.5 of Hu et al. (2019) yields that $\mathscr{F}$ has a unique minimizer $\mu^*$ in $\mathcal{P}$ and $\mu^*$ is absolutely continuous with respect to the Lebesgue measure. Moreover, $\mu^*$ satisfies the self-consistent condition: $\mu^*(X) \propto \exp\left(-\frac{1}{\lambda}\frac{\delta F(\mu^*)}{\delta\mu}(X)\right)$.

## 3.1 Proximal Gibbs Measure & Logarithmic Sobolev inequality

The aforementioned self-consistent relation motivates us to introduce the *proximal Gibbs distribution* (Nitanda et al., 2022; Chizat, 2022) whose density is given by

$$p_{\mathscr{X}}(X) \propto \exp\left(-\frac{1}{\lambda}\frac{\delta F(\mu_{\mathscr{X}})}{\delta\mu}(X)\right),$$

where $\mathscr{X} = (X^i)_{i=1}^N \subset \mathbb{R}^d$ is a set of $N$ particles and $X \in \mathbb{R}^d$. As we will see, the convergence of MFLD heavily depends on a *logarithmic Sobolev inequality* (LSI) on the proximal Gibbs measure.

**Definition 2** (Logarithmic Sobolev inequality). *Let $\mu$ be a probability measure on $(\mathbb{R}^d, \mathcal{B}(\mathbb{R}^d))$. $\mu$ satisfies the LSI with constant $\alpha > 0$ if for any smooth function $\phi : \mathbb{R}^d \to \mathbb{R}$ with $\mathbb{E}_\mu[\phi^2] < \infty$, we have $\mathbb{E}_\mu[\phi^2 \log(\phi^2)] - \mathbb{E}_\mu[\phi^2]\log(\mathbb{E}_\mu[\phi^2]) \leq \frac{2}{\alpha}\mathbb{E}_\mu[\|\nabla\phi\|_2^2]$.*

This is equivalent to the condition that the KL divergence from $\mu$ is bounded by the Fisher divergence: $\int \log(d\nu/d\mu)d\nu \leq \frac{1}{2\alpha}\int \|\nabla\log(d\nu/d\mu)\|^2 d\nu$, for any $\nu \in \mathcal{P}$ which is absolutely continuous with respect to $\mu$. Our analysis requires that the proximal Gibbs distribution satisfies the LSI as follows.

**Assumption 3.** *$\mu^*$ and $p_{\mathscr{X}}$ satisfy the LSI with $\alpha > 0$ for any set of particles $\mathscr{X} = (X^i)_{i=1}^N \subset \mathbb{R}^d$.*

**Verification of LSI.** The LSI of proximal Gibbs measure can be established via standard perturbation criteria. For $U(\mu)$ with bounded first-variation, we may apply the classical Bakry-Emery and Holley-Stroock arguments (Bakry and Émery, 1985a; Holley and Stroock, 1987) (see also Corollary 5.7.2 and 5.1.7 of Bakry et al. (2014)). Whereas for Lipschitz perturbations, we employ Miclo's trick (Bardet et al., 2018) or the more recent perturbation results in Cattiaux and Guillin (2022).

**Theorem 1.** *Under Assumptions 1 and 2, $\mu^*$ and $p_{\mathscr{X}}$ satisfy the log-Sobolev inequality with*

$$\alpha \geq \frac{\lambda_1}{2\lambda}\exp\left(-\frac{4R^2}{\lambda_1\lambda}\sqrt{2d/\pi}\right) \vee \left\{\frac{4\lambda}{\lambda_1} + e^{\frac{R^2}{2\lambda_1\lambda}}\left(\frac{R}{\lambda_1} + \sqrt{\frac{2\lambda}{\lambda_1}}\right)^2\left[2 + d + \frac{d}{2}\log\left(\frac{\lambda_2}{\lambda_1}\right) + 4\frac{R^2}{\lambda_1\lambda}\right]\right\}^{-1}.$$

*Furthermore, if $\|\frac{\delta U(\mu)}{\delta\mu}\|_\infty \leq R$ is satisfied for any $\mu \in \mathcal{P}_2$, then $\mu^*$ and $p_{\mathscr{X}}$ satisfy the LSI with $\alpha \geq \frac{\lambda_1}{\lambda}\exp\left(-\frac{4R}{\lambda}\right)$.*

See Lemma 5 for the proof of the first assertion, and Section A.1 of Nitanda et al. (2022) or Proposition 5.1 of Chizat (2022) for the proof of the second assertion.

# 4  Main Result: Convergence Analysis

In this section, we present our the convergence rate analysis of the discretized dynamics Eq. (6). To derive the convergence rate, we need to evaluate the errors induced by three approximations: (i) time discretization, (ii) particle approximation, and (iii) stochastic gradient.

## 4.1  General Recipe for Discretization Error Control

Note that for the finite-particle setting, the entropy term does not make sense because the negative entropy is not well-defined for a discrete empirical measure. Instead, we consider the distribution of $N$ particles, that is, let $\mu^{(N)} \in \mathcal{P}^{(N)}$ be a distribution of $N$ particles $\mathscr{X} = (X^i)_{i=1}^N$ where $\mathcal{P}^{(N)}$ is the set of probability measures on $(\mathbb{R}^{d \times N}, \mathcal{B}(\mathbb{R}^{d \times N}))$. Similarly, we introduce the following objective on $\mathcal{P}^{(N)}$:

$$\mathscr{F}^N(\mu^{(N)}) = N\mathbb{E}_{\mathscr{X} \sim \mu^{(N)}}[F(\mu_{\mathscr{X}})] + \lambda \mathrm{Ent}(\mu^{(N)}).$$

One can easily verify that if $\mu^{(N)}$ is a product measure of $\mu \in \mathcal{P}$, then $\mathscr{F}^N(\mu^{(N)}) \geq N\mathscr{F}(\mu)$ by the convexity of $F$. *Propagation of chaos* (Sznitman, 1991) refers to the phenomenon that, as the number of particles $N$ increases, the particles behave as if they are independent; in other words, the joint distribution of the $N$ particles becomes "close" to the product measure. Consequently, the minimum of $\mathscr{F}^N(\mu^{(N)})$ (which can be obtained by the particle-approximated MFLD) is close to $N\mathscr{F}(\mu^*)$. Specifically, it has been shown in Chen et al. (2022) that

$$0 \leq \inf_{\mu^{(N)} \in \mathcal{P}^{(N)}} \frac{1}{N}\mathscr{F}^N(\mu^{(N)}) - \mathscr{F}(\mu^*) \leq \frac{C_\lambda}{N}, \tag{7}$$

for some constant $C_\lambda > 0$ (see Section E for more details). Importantly, if we consider the Wasserstein gradient flow on $\mathcal{P}^{(N)}$, the convergence rate of which depends on the logarithmic Sobolev inequality (Assumption 3), we need to ensure that the LSI constant does not deteriorate as $N$ increases. Fortunately, the propagation of chaos and the *tensorization* of LSI entail that the LSI constant with respect to the objective $\mathscr{F}^N$ can be uniformly bounded over all choices of $N$ (see Eq. (11) in the appendix).

To deal with the time discretization error, we build upon the one-step interpolation argument from Vempala and Wibisono (2019) which analyzed the vanilla gradient Langevin dynamics (see also Nitanda et al. (2022) for its application to the infinite-particle MFLD).

Bounding the error induced by the stochastic gradient approximation is also challenging because the objective is defined on the space of probability measures, and thus techniques in finite-dimensional settings cannot be utilized in a straightforward manner. Roughly speaking, this error is characterized by the variance of $v_k^i$: $\sigma_{v,k}^2 := \mathbb{E}_{\omega_k, \mathscr{X}_{0:k}}[\|v_k^i - \nabla\frac{\delta F(\mu_k)}{\delta\mu}\|^2]$ [1]. In addition, to obtain a refined evaluation, we also incorporate the following smoothness assumption.

**Assumption 4.** $v_k^i(\mathscr{X}_{0:k}, \omega_k^i)$ and $D_m^i F(\mathscr{X}_k) = \nabla\frac{\delta F(\mu_{\mathscr{X}_k})}{\delta\mu}(X^i_k)$ are differentiable w.r.t. $(X_k^j)_{j=1}^N$ and, for either $G(\mathscr{X}_k) = v_k^i(\mathscr{X}_{0:k}, \omega_k^i)$ or $G(\mathscr{X}_k) = D_m^i F(\mathscr{X}_k)$ as a function of $\mathscr{X}_k$, it is satisfied that $\|G(\mathscr{X}_k') - G(\mathscr{X}_k) - \sum_{j=1}^N \nabla_{X_k^j}^\top G(\mathscr{X}_k) \cdot (X_k^{\prime j} - X_k^j)\| \leq Q\frac{\sum_{j \neq i}\|X_k^{\prime j} - X_k^j\|^2 + N\|X_k^{\prime i} - X_k^i\|^2}{2N}$ for some $Q > 0$ and $\mathscr{X}_k = (X_k^j)_{j=1}^N$ and $\mathscr{X}_k' = (X_k^{\prime j})_{j=1}^N$. We also assume $\mathbb{E}_{\omega_k}\big[\big\|\nabla_{X_k^j}v_k^i(\mathscr{X}_{0:k}, \omega_k^i)^\top - \nabla_{X_k^j}D_m^i F(\mathscr{X}_k)^\top\big\|_{\mathrm{op}}^2\big] \leq \frac{1+\delta_{i,j}N^2}{N^2}\tilde{\sigma}_{v,k}^2$, and $\|v_k^i((\mathscr{X}_k, \mathscr{X}_{0:k-1}), \omega_k^i) - v_k^i((\mathscr{X}_k', \mathscr{X}_{0:k-1}), \omega_k^i)\| \leq L(W_2(\mu_{\mathscr{X}_k}, \mu_{\mathscr{X}_k'}) + \|X_k^i - X_k^{\prime i}\|)$.

Note that this assumption is satisfied if the gradient is twice differentiable and the second derivative is bounded. The factor $N$ appears because the contribution of each particle is $O(1/N)$ unless $i = j$. In the following, we present both the basic convergence result under Assumption 2, and the improved rate under the additional Assumption 4.

---

[1] After the acceptance of this manuscript, we noticed that by replacing $\sigma_{v,k}^2$ with the square of a conditional expectation $\mathbb{E}_{\mathscr{X}_{0:k+1}}[\|\mathbb{E}_{\omega_k|\mathscr{X}_{0:k+1}}[v_k^i - \nabla\frac{\delta F(\mu_k)}{\delta\mu}]\|^2]$, we can obtain a refined bound as in Das et al. (2023) suggested by anonymous reviewer Y8on. We defer such refined analysis to future work.

Taking these factors into consideration, we can evaluate the decrease in the objective value after one-step update. Let $\mu_k^{(N)} \in \mathcal{P}^{(N)}$ be the distribution of $\mathscr{X}_k$ conditioned on the sequence $\omega_{0:k} = (\omega_{k'})_{k'=0}^{k}$. Define

$$\bar{R}^2 := \mathbb{E}[\|X_0^i\|^2] + \frac{1}{\lambda_1}\left[\left(\frac{\lambda_1}{4\lambda_2} + \frac{1}{\lambda_1}\right)(R^2 + \lambda_2 c_r) + \lambda d\right], \quad \delta_\eta := C_1 \bar{L}^2(\eta^2 + \lambda\eta),$$

where $C_1 = 8[R^2 + \lambda_2(c_r + \bar{R}^2) + d]$ and $\bar{L} = L + \lambda_2$.[2]

**Theorem 2.** *Under Assumptions 1, 2 and 3, if $\eta_k \leq \lambda_1/(4\lambda_2)$, we have*

$$\frac{1}{N}\mathbb{E}\left[\mathscr{F}^N(\mu_{k+1}^{(N)})\right] - \mathscr{F}(\mu^*) \leq \exp\left(-\frac{\lambda\alpha\eta_k}{2}\right)\left(\frac{1}{N}\mathbb{E}\left[\mathscr{F}^N(\mu_k^{(N)})\right] - \mathscr{F}(\mu^*)\right) + \eta_k\left(\delta_{\eta_k} + \frac{C_\lambda}{N}\right) + \Upsilon_k,$$

*where $\Upsilon_k = 4\eta_k\delta_{\eta_k} + (R + \lambda_2\bar{R})(1 + \sqrt{\lambda/\eta_k})\left(\eta_k^2\tilde{\sigma}_{v,k}\sigma_{v,k} + Q\eta_k^3\sigma_{v,k}^2\right) + \eta_k^2(L + \lambda_2)^2\sigma_{v,k}^2$ with Assumption 4, and $\Upsilon_k = \sigma_{v,k}^2\eta_k$ without this additional assumption; the expectation is taken with respect to the randomness $(\omega_{k'})_{k'=1}^{k} = ((\omega_{k'}^i)_{i=1}^{N})_{k'=1}^{k}$ of the stochastic gradient; and $C_\lambda = 2\lambda L\alpha(1 + 2c_L\bar{R}^2) + 2\lambda^2 L^2\bar{R}^2$.*

The proof can be found in Appendix B. We remark that to derive the bound for $\Upsilon_k = \sigma_{v,k}^2\eta_k$ is relatively straightforward, but to derive a tighter bound with Assumption 4 is technically challenging, because we need to evaluate how the next-step distribution $\mu_{k+1}^{(N)}$ is correlated with the stochastic gradient $v_k^i$. Evaluating such correlations is non-trivial because the randomness is induced not only by $\omega_k$ but also by the Gaussian noise. Thanks to this general result, we only need to evaluate the variance $\sigma_{v,k}$ and $\tilde{\sigma}_{v,k}$ for each method to obtain the specific convergence rate.

**Conversion to a Wasserstein distance bound.** As a consequence of the bound on $\frac{1}{N}\mathscr{F}^N(\mu_k^{(N)}) - \mathscr{F}(\mu^*)$, we can control the Wasserstein distance between $\mu_k^{(N)}$ and $\mu^{*N}$, where $\mu^{*N} \in \mathcal{P}^{(N)}$ is the ($N$-times) product measure of $\mu^*$. Let $W_2(\mu, \nu)$ be the 2-Wasserstein distance between $\mu$ and $\nu$, then

$$W_2^2(\mu_k^{(N)}, \mu^{*N}) \leq \frac{2}{\lambda\alpha}(\mathscr{F}^N(\mu_k^{(N)}) - N\mathscr{F}(\mu^*)),$$

under Assumptions 1 and 3 (see Lemma 3 in the Appendix). Hence, if $\frac{1}{N}\mathscr{F}^N(\mu_k^{(N)}) - \mathscr{F}(\mu^*)$ is small, the particles $(X_k^i)_{i=1}^{N}$ behaves like an i.i.d. sample from $\mu^*$. As an example, for the mean-field neural network setting, if $|h_x(z) - h_{x'}(z)| \leq L\|x - x'\|$ ($\forall x, x' \in \mathbb{R}^d$) and $V_{\mu^*} = \int(f_{\mu^*}(z) - h_x(z))^2 \mathrm{d}\mu^*(x) < \infty$ for a fixed $z$, then Lemma 4 in the appendix yields that

$$\mathbb{E}_{\mathscr{X}_k \sim \mu_k^{(N)}}[(f_{\mu_{\mathscr{X}_k}}(z) - f_{\mu^*}(z))^2] \leq \frac{2L^2}{N}W_2^2(\mu_k^{(N)}, \mu^{*N}) + \frac{2}{N}V_{\mu^*},$$

which also gives $\mathbb{E}_{\mathscr{X}_k \sim \mu_k^{(N)}}[(f_{\mu_{\mathscr{X}_k}}(z) - f_{\mu^*}(z))^2] \leq \frac{4L^2}{\lambda\alpha}\left(N^{-1}\mathscr{F}^N(\mu_k^{(N)}) - \mathscr{F}(\mu^*)\right) + \frac{2}{N}V_{\mu^*}$. This allows us to monitor the convergence of the finite-width neural network to the optimal solution $\mu^*$ in terms of the model output (up to $1/N$ error).

## 4.2 F-MFLD and SGD-MFLD

Here, we present the convergence rate for F-MFLD and SGD-MFLD simultaneously. F-MFLD can be seen as a special case of SGD-MFLD where the variance $\sigma_{v,k}^2 = 0$. We specialize the previous assumptions to the stochastic gradient setting as follows.

**Assumption 5.**

(i) $\sup_{x\in\mathbb{R}^d}\|\nabla\frac{\delta\ell(\mu,z)}{\delta\mu}(x)\| \leq R$ for all $\mu \in \mathcal{P}$ and $z \in \mathcal{Z}$.

(ii) $\sup_x\|\nabla_x\nabla_x^\top\frac{\delta\ell(\mu,z)}{\delta\mu}(x)\|_{\mathrm{op}}$, $\sup_{x,x'}\|\nabla_x\nabla_{x'}^\top\frac{\delta^2\ell(\mu,z)}{\delta^2\mu}(x,x')\|_{\mathrm{op}} \leq R$.

Note that point $(i)$ is required to bound the variance $\sigma_{v,k}^2$ (i.e., $\sigma_{v,k}^2 \leq R^2/B$), whereas point $(ii)$ corresponds to the additional Assumption 4 required for the improved convergence rate. Let $\Delta_0 := \frac{1}{N}\mathbb{E}[\mathscr{F}^N(\mu_0^{(N)})] - \mathscr{F}(\mu^*)$. We have the following evaluation of the objective. Note that we can recover the evaluation for F-MFLD by formally setting $B = \infty$ so that $\sigma_{v,k}^2 = 0$ and $\tilde{\sigma}_{v,k}^2 = 0$.

---

[2]The constants are not optimized (e.g., $O(\frac{1}{\lambda_1})$ in $\bar{R}$ can be improved) since we prioritize for clean presentation.

**Theorem 3.** *Suppose that $\eta_k = \eta \ (\forall k \in \mathbb{N}_0)$ and $\lambda\alpha\eta \leq 1/4$ and $\eta \leq \lambda_1/(4\lambda_2)$. Under Assumptions 1, 2, 3 and 5, it holds that*

$$\frac{1}{N}\mathbb{E}[\mathscr{F}^N(\mu_k^{(N)})] - \mathscr{F}(\mu^*) \leq \exp\left(-\lambda\alpha\eta k/2\right)\Delta_0 + \frac{4}{\lambda\alpha}\bar{L}^2 C_1\left(\lambda\eta + \eta^2\right) + \frac{4}{\lambda\alpha\eta}\bar{\Upsilon} + \frac{4C_\lambda}{\lambda\alpha N}, \quad (8)$$

*where $\bar{\Upsilon} = 4\eta\delta_\eta + \left[R + \lambda_2\bar{R} + (L + \lambda_2)^2\right](1 + \sqrt{\frac{\lambda}{\eta}})\eta^2\frac{R^2}{B} + (R + \lambda_2\bar{R})R(1 + \sqrt{\frac{\lambda}{\eta}})\eta^3\frac{R^2}{B}$ under Assumption 5-(ii), and $\bar{\Upsilon} = \frac{R^2}{B}\eta$ without this additional assumption.*

The proof is given in Appendix C. This can be seen as a mean-field generalization of Vempala and Wibisono (2019) which provides a convergence rate of discrete time vanilla GLD with respect to the KL divergence. Indeed, their derived rate $O(\exp(-\lambda\alpha\eta k)\Delta_0 + \frac{\eta}{\alpha\lambda})$ is consistent to ours, since Eq. (26) in the Appendix implies that the objective $\frac{1}{\lambda}(\frac{1}{N}\mathbb{E}[\mathscr{F}^N(\mu_k^{(N)})] - \mathscr{F}(\mu^*))$ upper bounds the KL divergence between $\mu_k^{(N)}$ and $\mu^{*N}$. Moreover, our result also handles the stochastic approximation which can give better total computational complexity even in the vanilla GLD setting as shown below.

For a given $\epsilon > 0$, if we take

$$\eta \leq \frac{\alpha\epsilon}{40\bar{L}^2 C_1} \wedge \frac{1}{\bar{L}}\sqrt{\frac{\lambda\alpha\epsilon}{40C_1}} \wedge 1, \quad k \geq \frac{2}{\lambda\alpha\eta}\log(2\Delta_0/\epsilon),$$

with $B \geq 4\left[(1 + R)(R + \lambda_2\bar{R}) + (L + \lambda_2)^2\right]R^2(\eta + \sqrt{\eta\lambda})/(\lambda\alpha\epsilon)$ under Assumption 5-(ii) and $B \geq 4R^2/(\lambda\alpha\epsilon)$ without such assumption, then the right hand side of (8) can be bounded as $\frac{1}{N}\mathbb{E}[\mathscr{F}^N(\mu_k^{(N)})] - \mathscr{F}(\mu^*) = \epsilon + \frac{4C_\lambda}{\lambda\alpha N}$. Hence we achieve $\epsilon + O(1/N)$ error with iteration complexity:

$$k = O\left(\frac{\bar{L}^2}{\alpha\epsilon} + \frac{\bar{L}}{\sqrt{\lambda\alpha\epsilon}}\right)\frac{1}{\lambda\alpha}\log(\epsilon^{-1}). \quad (9)$$

If we neglect $O(1/\sqrt{\lambda\alpha\epsilon})$ as a second order term, then the above can be simplified as $O\left(\frac{\bar{L}^2}{\lambda\alpha^2}\frac{\log(\epsilon^{-1})}{\epsilon}\right)$.

Noticeably, the mini-batch size $B$ can be significantly reduced under the additional smoothness condition Assumption 5-(ii) (indeed, we have $O(\eta + \sqrt{\eta\lambda})$ factor reduction). Recall that when the objective is an empirical risk, the gradient complexity per iteration required by F-MFLD is $O(n)$. Hence, the total complexity of F-MFLD is $O(nk)$ where $k$ is given in Eq. (9). Comparing this with SGD-MFLD with mini-batch size $B$, we see that SGD-MFLD has better total computational complexity ($Bk$) when $n > B$. In particular, if $B = \Omega((\eta + \sqrt{\eta\lambda})/(\lambda\alpha\epsilon)) \geq \Omega(\lambda^{-1} + \sqrt{(\epsilon\lambda\alpha)^{-1}})$ which yields the objective value of order $O(\epsilon)$, SGD-MFLD achieves $O(n/B) = O(n(\lambda \wedge \sqrt{\epsilon\lambda\alpha}))$ times smaller total complexity. For example, when $\lambda = \alpha\epsilon = 1/\sqrt{n}$, a $O(\sqrt{n})$-factor reduction of total complexity can be achieved by SGD-MFLD, which is a significant improvement.

### 4.3 SVRG-MFLD

Now we present the convergence rate of SVRG-MFLD in the fixed step size setting where $\eta_k = \eta \ (\forall k)$. Instead of Assumption 5, we introduce the following two assumptions.

**Assumption 6.**

(i) *For any $i \in [n]$, it holds that $\left\|\nabla\frac{\delta\ell_i(\mu)}{\delta\mu}(x) - \nabla\frac{\delta\ell_i(\mu')}{\delta\mu}(x')\right\| \leq L(W_2(\mu, \mu') + \|x - x'\|)$ for any $\mu, \mu' \in \mathcal{P}_2$ and $x, x' \in \mathbb{R}^d$, and $\sup_{x \in \mathbb{R}^d}\|\nabla\frac{\delta\ell_i(\mu)}{\delta\mu}(x)\| \leq R$ for any $\mu \in \mathcal{P}$.*

(ii) *Additionally, we have $\sup_x \|\nabla_x\nabla_x^\top\frac{\delta\ell_i(\mu)}{\delta\mu}(x)\|_{op}, \sup_{x,x'}\|\nabla_x\nabla_{x'}^\top\frac{\delta^2\ell_i(\mu)}{\delta^2\mu}(x, x')\|_{op} \leq R$.*

Here again, point (i) is required to bound $\sigma_{v,k}^2$ and point (ii) yields Assumption 4. We have the following computational complexity bound for SVRG-MFLD.

**Theorem 4.** *Suppose that $\lambda\alpha\eta \leq 1/4$ and $\eta \leq \lambda_1/(4\lambda_2)$. Let $\Xi = \frac{n-B}{B(n-1)}$. Then, under Assumptions 1, 2, 3 and 6, we have the same error bound as Eq. (8) with different $\bar{\Upsilon}$:*

$$\bar{\Upsilon} = 4\eta\delta_\eta + \left(1 + \sqrt{\frac{\lambda}{\eta}}\right)\left\{(R + \lambda_2\bar{R})\eta^2\sqrt{C_1\Xi L^2 m(\eta^2 + \eta\lambda)R^2\Xi} + \left[(R + \lambda_2\bar{R})R\eta^3 + (L + \lambda_2)^2\eta^2\right]\right\},$$

*under Assumption 6-(ii), and $\bar{\Upsilon} = C_1\Xi L^2 m\eta^2(\eta + \lambda)$ without Assumption 6-(ii).*

The proof is given in Appendix D. Therefore, to achieve $\frac{1}{N}\mathbb{E}[\mathscr{F}^N(\mu_k^{(N)})] - \mathscr{F}(\mu^*) \leq O(\epsilon) + \frac{4C_\lambda}{\lambda\alpha N}$ for a given $\epsilon > 0$, it suffices to set

$$\eta = \frac{\alpha\epsilon}{40\bar{L}^2 C_1} \wedge \frac{\sqrt{\lambda\alpha\epsilon}}{40\bar{L}\sqrt{C_1}}, \quad k = \frac{2\log(2\Delta_0/\epsilon)}{\lambda\alpha\eta} = O\left(\frac{\bar{L}^2}{\alpha\epsilon} + \frac{\bar{L}}{\sqrt{\lambda\alpha\epsilon}}\right)\frac{\log(\epsilon^{-1})}{(\lambda\alpha)},$$

with $B \geq \left[\sqrt{m}\frac{(\eta+\sqrt{\eta/\lambda})^2}{\eta\lambda} \vee m\frac{(\eta+\sqrt{\eta/\lambda})^3}{\eta\lambda}\right] \wedge n$. In this setting, the total gradient complexity can be bounded as

$$Bk + \frac{nk}{m} + n \lesssim \max\left\{n^{\frac{1}{3}}\left(1 + \sqrt{\frac{\eta}{\lambda}}\right)^{\frac{4}{3}}, \sqrt{n}(\eta\lambda)^{\frac{1}{4}}\left(1 + \sqrt{\frac{\eta}{\lambda}}\right)^{\frac{3}{2}}\right\}\frac{1}{\eta}\frac{\log(\epsilon^{-1})}{\lambda\alpha} + n,$$

where $m = \Omega(n/B) = \Omega([n^{2/3}(1 + \sqrt{\eta/\lambda})^{-4/3} \wedge \sqrt{n}(1 + \sqrt{\eta/\lambda})^{-3/2}(\sqrt{\eta\lambda})^{-1/2})] \vee 1)$ and $B = O\left(\left[n^{\frac{1}{3}}\left(1 + \sqrt{\frac{\eta}{\lambda}}\right)^{\frac{4}{3}} \vee \sqrt{n}(\eta\lambda)^{\frac{1}{4}}\left(1 + \sqrt{\frac{\eta}{\lambda}}\right)^{\frac{3}{2}}\right] \wedge n\right)$. When $\epsilon$ is small, the first term in the right hand side becomes the main term which is $\max\{n^{1/3}, \sqrt{n}(\alpha\epsilon\lambda)^{1/4}\}\frac{\log(\epsilon^{-1})}{\lambda\alpha^2\epsilon}$. Therefore, comparing with the full batch gradient method (F-MFLD), SVRG-MFLD achieves at least $\min\left\{n^{\frac{2}{3}}, \sqrt{n}(\alpha\epsilon\lambda)^{-\frac{1}{4}}\right\}$ times better total computational complexity when $\eta \leq \lambda$. Note that even without the additional Assumption 6-(ii), we still obtain a $\sqrt{n}$-factor improvement (see Appendix D). This indicates that variance reduction is indeed effective to improve the computational complexity, especially in a large sample size setting.

Finally, we compare the convergence rate in Theorem 4 (setting $F$ to be linear) against prior analysis of standard gradient Langevin dynamics (LD) under LSI. To our knowledge, the current best convergence rate of LD in terms of KL divergence was given in Kinoshita and Suzuki (2022), which yields a $O\left(\left(n + \frac{\sqrt{n}}{\epsilon}\right)\frac{\log(\epsilon^{-1})}{(\lambda\alpha)^2}\right)$ iteration complexity to achieve $\epsilon$ error. This corresponds to our analysis under Assumption 6-(i) (without (ii)) (see Appendix D). Note that in this setting, our bound recovers their rate even though our analysis is generalized to nonlinear mean-field functionals (the $O(\lambda)$-factor discrepancy is due to the difference in the objective – their bound considers the KL divergence while our objective corresponds to $\lambda$ times the KL divergence). Furthermore, our analysis gives an even faster convergence rate under an additional mild assumption (Assumption 6-(ii)).

## 5  Conclusion

We gave a unified theoretical framework to bound the optimization error of the *single-loop* mean-field Langevin dynamics (MFLD) that is applicable to the finite-particle, discrete-time, and stochastic gradient algorithm. Our analysis is general enough to cover several important learning problems such as the optimization of mean-field neural networks, density estimation via MMD minimization, and variational inference via KSD minimization. We considered three versions of the algorithms (F-MFLD, SGD-MFLD, SGLD-MFLD); and despite the fact that our analysis deals with a more general setting (mean-field interactions), we are able to recover and even improve existing convergence guarantees when specialized to the standard gradient Langevin dynamics.

## Acknowledgements

TS was partially supported by JSPS KAKENHI (20H00576) and JST CREST (JPMJCR2115). AN was partially supported by JSPS KAKENHI (22H03650). We thank the anonymous reviewers for their careful reading of our manuscript and their insightful comments.

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

**Table of Contents**

## A  Verification of Assumptions

Assumption 2 can be satisfied in the following settings for the three examples presented in Section 2.1.

(i) **mean-field neural network.** The neurons $h_x(\cdot)$ and their gradients are bounded (e.g., tanh activation), and the first derivative of the loss is Lipschitz continuous (e.g., squared loss, logistic loss). That is, there exists $C > 0$ such that $\sup_z \sup_x |h_x(z)| \leq C$, $\sup_z \sup_x \|\nabla_x h_x(z)\| \leq C$ and $\|\nabla h(x) - \nabla h(x')\| \leq C\|x - x'\|$ for all $x, x' \in \mathbb{R}^d$, and $\sup_{y,z} \sup_{\mu \in \mathcal{P}} |\partial_f \ell(f_\mu(z), y)| \leq C$, $|\partial_f \ell(f_\mu(z), y) - \partial_f \ell(f_{\mu'}(z), y)| \leq C|f_\mu(z) - f_{\mu'}(z)|$ and $|\partial_f^2 \ell(f_\mu(z), y)| \leq C$ for all $(y, z)$ and $\mu, \mu' \in \mathcal{P}$.

(ii) **MMD minimization.** The kernel $k$ is smooth and has light tail, e.g., the Gaussian RBF kernel.

(iii) **KSD minimization.** The kernel $k$ has a light tail such that $\sup_{z,z'} \max\{|W_{\mu^*}(z, z')|,$ $\|\nabla_z W_{\mu^*}(z, z')\|, \|\nabla_z \nabla_z^\top W_{\mu^*}(z, z')\|_{\mathrm{op}}\} \leq C$; for example, $k(z, z') = \exp\big(-\frac{\|z\|^2}{2\sigma_1^2} - \frac{\|z'\|^2}{2\sigma_1^2} - \frac{\|z-z'\|^2}{2\sigma_2^2}\big)$, and $\max\{\|\nabla \log(\mu^*(z))\|, \|\nabla^{\otimes 2} \log(\mu^*(z))\|_{\mathrm{op}}, \|\nabla^{\otimes 3} \log(\mu^*(z))\|_{\mathrm{op}}\} \leq C(1 + \|z\|)$.

## B  Proof of Theorem 2

This section gives the proof of Theorem 2. For notation simplicity, we write $\eta$ to indicate $\eta_k$. Recall that for $\mathscr{X} = (X^i)_{i=1}^N$, the proximal Gibbs distribution $\mu_\mathscr{X}$ is defined by

$$p_{\mathscr{X}'}(X) \propto \exp\left(-\frac{1}{\lambda}\frac{\delta F(\mu_{\mathscr{X}'})}{\delta\mu}(X)\right).$$

We also define another version of the proximal Gibbs distribution corresponding to $\mathscr{F}^N$ as

$$p^{(N)}(\mathscr{X}) \propto \exp\left(-\frac{N}{\lambda}F(\mu_\mathscr{X})\right).$$

### B.1  Evaluation of Objective Decrease

By the same argument as Chen et al. (2022), we have that

$$-\frac{1}{\lambda}\nabla\frac{\delta F(\mu_\mathscr{X})}{\delta\mu}(X^i) = \nabla\log(p_\mathscr{X}(X_i)) = \nabla_i\log(p^{(N)}(\mathscr{X})) = -\frac{N}{\lambda}\nabla_i F(\mu_\mathscr{X}),$$

where $\nabla_i$ is the partial derivative with respect to $X_i$. Therefore, we have

$$\begin{aligned}
\nabla_i\frac{\delta\mathscr{F}^N(\mu^{(N)})}{\delta\mu}(\mathscr{X}) &= N\nabla_i F(\mu_\mathscr{X}) + \lambda\nabla_i\log(\mu^{(N)}(\mathscr{X})) \\
&= \nabla\frac{\delta F(\mu_\mathscr{X})}{\delta\mu}(X_i) + \lambda\nabla_i\log(\mu^{(N)}(\mathscr{X})) \\
&= -\lambda\nabla_i\log(p^{(N)}(\mathscr{X})) + \lambda\nabla_i\log(\mu^{(N)}(\mathscr{X})).
\end{aligned} \tag{10}$$

Remembering that $\mathscr{X}_k = (X_k^i)_{i=1}^N$ is updated by $X_{k+1}^i = X_k^i - \eta v_k^i + \sqrt{2\lambda\eta}\xi_k^i$, the solutions $X_k^i$ and $X_{k+1}^i$ can be interpolated by the following continuous time dynamics:

$$\begin{aligned}
\widetilde{\mathscr{X}_0} &= \mathscr{X}_k, \\
\mathrm{d}\widetilde{X}_t^i &= -v_k^i\mathrm{d}t + \sqrt{2\lambda}\mathrm{d}W_t^i,
\end{aligned}$$

for $0 \leq t \leq \eta$. Then, $\widetilde{\mathscr{X}_\eta}$ obeys the same distribution as $\mathscr{X}_{k+1}$. Let $\tilde{\mu}_t^{(N)}$ be the law of $\widetilde{\mathscr{X}_t}$. The Fokker-Planck equation of the dynamics yields that

$$\frac{\mathrm{d}\tilde{\mu}_t^{(N)}}{\mathrm{d}t}(\mathscr{X}|\mathscr{X}_{0:k}, \omega_{0:k}) = \sum_{i=1}^N \nabla_i\cdot\left(\tilde{\mu}_t^{(N)}(\mathscr{X}|\mathscr{X}_{0:k}, \omega_{0:k})v_k^i\right) + \lambda\sum_{i=1}^n \Delta_i\tilde{\mu}_t^{(N)}(\mathscr{X}|\mathscr{X}_{0:k}, \omega_{0:k}).$$

Hence, by taking expectation with respect to $\mathscr{X}_{0:k}$, it holds that

$$
\frac{\mathrm{d}\tilde{\mu}_t^{(N)}}{\mathrm{d}t}(\mathscr{X})
$$

$$
= \sum_{i=1}^N \lambda \nabla_i \cdot \left( \tilde{\mu}_t^{(N)}(\mathscr{X}) \left( \nabla_i \log \left( \frac{\tilde{\mu}_t^{(N)}}{p^{(N)}}(\mathscr{X}) \right) \right) \right)
$$

$$
+ \sum_{i=1}^N \nabla_i \cdot \left\{ \tilde{\mu}_t^{(N)}(\mathscr{X}) \left( \mathbb{E}_{\mathscr{X}_{0:k}|\widetilde{\mathscr{X}_t}} \left[ v_k^i(\mathscr{X}_{0:k}, \omega_k^i) \ \Big| \ \widetilde{\mathscr{X}_t} = \mathscr{X}, \omega_k \right] - \nabla \frac{\delta F(\mu_{\widetilde{\mathscr{X}_t}})}{\delta \mu}(X^i) \right) \right\},
$$

where we omitted the influence of $\omega_{0:k}$ to $\tilde{\mu}_t^{(N)}$, which should be written as $\tilde{\mu}_t^{(N)}(\mathscr{X}|\omega_{0:k})$ in a more precise manner. Combining this and Eq. (10) yields the following decomposition,

$$
\frac{\mathrm{d}\mathscr{F}^N(\tilde{\mu}_t^{(N)})}{\mathrm{d}t}
$$

$$
\leq - \underbrace{\frac{3\lambda^2}{4} \sum_{i=1}^N \mathbb{E}_{\mathscr{X} \sim \tilde{\mu}_t^{(N)}} \left[ \left\| \nabla_i \log \left( \frac{\tilde{\mu}_t^{(N)}}{p^{(N)}}(\mathscr{X}) \right) \right\|^2 \right]}_{=:A}
$$

$$
+ \underbrace{\sum_{i=1}^N \mathbb{E}_{\widetilde{\mathscr{X}_t}, \widetilde{\mathscr{X}_0}} \left[ \left\| \nabla_i \frac{\delta F(\tilde{\mu}_0^{(N)})}{\delta \mu}(\widetilde{X}_0^i) - \nabla_i \frac{\delta F(\tilde{\mu}_t^{(N)})}{\delta \mu}(\widetilde{X}_t^i) \right\|^2 \right]}_{=:B}
$$

$$
- \underbrace{\sum_{i=1}^N \lambda \mathbb{E}_{\widetilde{\mathscr{X}_t}, \widetilde{\mathscr{X}_0}} \left[ \left\langle \nabla_i \log \left( \frac{\tilde{\mu}_t^{(N)}}{p^{(N)}}(\widetilde{\mathscr{X}_t}) \right), v_k^i(\mathscr{X}_{0:k}, \omega_k^i) - \nabla \frac{\delta F(\mu_{\widetilde{\mathscr{X}_0}})}{\delta \mu}(\widetilde{X}_0^i) \right\rangle \right]}_{=:C}.
$$

**Evaluation of term $A$:** By the *leave-one-out argument* in the proof of Theorem 2.1 of Chen et al. (2022), the first term of the right hand side can be upper bounded by

$$
- \frac{3\lambda^2}{4N} \sum_{i=1}^N \mathbb{E}_{\mathscr{X} \sim \tilde{\mu}_t^{(N)}} \left[ \left\| \nabla_i \log \left( \frac{\tilde{\mu}_t^{(N)}}{p^{(N)}}(\mathscr{X}) \right) \right\|^2 \right]
$$

$$
\leq - \frac{\lambda^2}{4N} \sum_{i=1}^N \mathbb{E}_{\mathscr{X} \sim \tilde{\mu}_t^{(N)}} \left[ \left\| \nabla_i \log \left( \frac{\tilde{\mu}_t^{(N)}}{p^{(N)}}(\mathscr{X}) \right) \right\|^2 \right]
$$

$$
- \frac{\lambda\alpha}{2} \left( \frac{1}{N} \mathscr{F}^N(\tilde{\mu}_t^{(N)}) - \mathscr{F}(\mu^*) \right) + \frac{C_\lambda}{N}, \tag{11}
$$

with a constant $C_\lambda$. The proof is given in Lemma 7 for completeness, where we see that the inequality holds with $C_\lambda = 2\lambda L \alpha (1 + 2c_L \bar{R}^2) + 2\lambda^2 L^2 \bar{R}^2$.

**Evaluation of term $B$:** By Lemma 2, we have

$$
\mathbb{E}_{\widetilde{\mathscr{X}_t}, \widetilde{\mathscr{X}_0}} \left[ \left\| \nabla \frac{\delta F(\tilde{\mu}_0^{(N)})}{\delta \mu}(\widetilde{X}_0^i) - \nabla \frac{\delta F(\tilde{\mu}_t^{(N)})}{\delta \mu}(\widetilde{X}_t^i) \right\|^2 \right] \leq \delta_{\eta_k},
$$

for $\delta_{\eta_k} = C_1 L^2(\eta_k^2 + \lambda \eta_k) = O(L^2(\eta_k^2 + \eta_k \lambda))$.

## B.2  Stochastic Gradient Error (Term $C$)

Now we evaluate the final term C. First, we derive a bound without Assumption 4. By the Cauchy-Schwarz inequality, we have that

$$
\lambda \mathbb{E}_{\widetilde{\mathscr{X}_t}, \widetilde{\mathscr{X}_0}} \left[ \left\langle \nabla_i \log \left( \frac{\tilde{\mu}_t^{(N)}}{p^{(N)}}(\widetilde{\mathscr{X}_t}) \right), v_k^i(\mathscr{X}_{0:k}, \omega_k^i) - \nabla \frac{\delta F(\mu_{\widetilde{\mathscr{X}_0}})}{\delta \mu}(\widetilde{X}_0^i) \right\rangle \right]
$$

$$\leq \frac{\lambda^2}{4} \mathbb{E}_{\widetilde{\mathscr{X}}_t, \widetilde{\mathscr{X}}_0} \left[ \left\| \nabla_i \log \left( \frac{\tilde{\mu}_t^{(N)}}{p^{(N)}}(\widetilde{\mathscr{X}}_t) \right) \right\|^2 \right] + \mathbb{E}_{\widetilde{\mathscr{X}}_t, \widetilde{\mathscr{X}}_0} \left[ \left\| v_k^i(\mathscr{X}_{0:k}, \omega_k^i) - \nabla \frac{\delta F(\mu_{\widetilde{\mathscr{X}}_0})}{\delta \mu}(\tilde{X}_0^i) \right\|^2 \right]$$

$$\leq \frac{\lambda^2}{4} \mathbb{E}_{\widetilde{\mathscr{X}}_t, \widetilde{\mathscr{X}}_0} \left[ \left\| \nabla_i \log \left( \frac{\tilde{\mu}_t^{(N)}}{p^{(N)}}(\widetilde{\mathscr{X}}_t) \right) \right\|^2 \right] + \sigma_{v,k}^2.$$

### B.2.1 Analysis under Assumption 4

Next, we derive a tighter result under the additional Assumption 4. We decompose term $C$ as follows:

$$\mathbb{E}_{\widetilde{\mathscr{X}}_t, \widetilde{\mathscr{X}}_0} \left[ \left\langle \nabla_i \log \left( \frac{\tilde{\mu}_t^{(N)}}{p^{(N)}}(\widetilde{\mathscr{X}}_t) \right), v_k^i(\mathscr{X}_{0:k}, \omega_k^i) - \nabla \frac{\delta F(\mu_{\widetilde{\mathscr{X}}_0})}{\delta \mu}(\tilde{X}_0^i) \right\rangle \right]$$

$$= \mathbb{E}_{\widetilde{\mathscr{X}}_t, \widetilde{\mathscr{X}}_0} \left[ \left\langle \nabla_i \log \left( \tilde{\mu}_t^{(N)}(\widetilde{\mathscr{X}}_t) \right), v_k^i(\mathscr{X}_{0:k}, \omega_k^i) - \nabla \frac{\delta F(\mu_{\widetilde{\mathscr{X}}_0})}{\delta \mu}(\tilde{X}_0^i) \right\rangle \right]$$

$$- \mathbb{E}_{\widetilde{\mathscr{X}}_t, \widetilde{\mathscr{X}}_0} \left[ \left\langle \nabla_i \log \left( p^{(N)}(\widetilde{\mathscr{X}}_t) \right), v_k^i(\mathscr{X}_{0:k}, \omega_k^i) - \nabla \frac{\delta F(\mu_{\widetilde{\mathscr{X}}_0})}{\delta \mu}(\tilde{X}_0^i) \right\rangle \right].$$

Since $\nabla_i \log(\tilde{\mu}_t^{(N)}) = \nabla_i \tilde{\mu}_t^{(N)} / \tilde{\mu}_t^{(N)}$, this is equivalent to

$$\mathbb{E}_{\widetilde{\mathscr{X}}_t, \widetilde{\mathscr{X}}_0} \left[ \left\langle \left( \tilde{\mu}_t^{(N)}(\widetilde{\mathscr{X}}_t)^{-1} \nabla_i \tilde{\mu}_t^{(N)}(\widetilde{\mathscr{X}}_t) - \nabla_i \log(p^{(N)})(\widetilde{\mathscr{X}}_t) \right), v_k^i(\mathscr{X}_{0:k}, \omega_k^i) - \nabla \frac{\delta F(\mu_{\widetilde{\mathscr{X}}_0})}{\delta \mu}(\tilde{X}_0^i) \right\rangle \right]$$

$$= \underbrace{\int \mathbb{E}_{\widetilde{\mathscr{X}}_0 | \widetilde{\mathscr{X}}_t} \left[ \left\langle \int \nabla_i \tilde{\mu}_t^{(N)}(\widetilde{\mathscr{X}}_t | \mathscr{X}_{0:k}') \mu^{(N)}(\mathrm{d}\mathscr{X}_{0:k}'), v_k^i(\mathscr{X}_{0:k}, \omega_k^i) - \nabla \frac{\delta F(\mu_{\mathscr{X}_k})}{\delta \mu}(X_k^i) \right\rangle \right] \mathrm{d}\widetilde{\mathscr{X}}_t}_{C\text{-(I)}}$$

$$- \underbrace{\mathbb{E}_{\widetilde{\mathscr{X}}_0, \widetilde{\mathscr{X}}_t} \left[ \left\langle \nabla_i \log(p^{(N)})(\widetilde{\mathscr{X}}_t), v_k^i(\mathscr{X}_{0:k}, \omega_k^i) - \nabla \frac{\delta F(\mu_{\widetilde{\mathscr{X}}_0})}{\delta \mu}(\tilde{X}_0^i) \right\rangle \right]}_{C\text{-(II)}}, \tag{12}$$

where $\mathscr{X}_{0:k}'$ is an independent copy of $\mathscr{X}_{0:k}$.

*Evaluation of C-(I):*
(1) The first term ($C$-(I)) of the right hand side of Eq. (12) can be evaluated as

$$\int \mathbb{E}_{\widetilde{\mathscr{X}}_0 | \widetilde{\mathscr{X}}_t} \left[ \left\langle \int \nabla_i \tilde{\mu}_t^{(N)}(\widetilde{\mathscr{X}}_t | \mathscr{X}_{0:k}') \mu^{(N)}(\mathrm{d}\mathscr{X}_{0:k}'), v_k^i(\mathscr{X}_{0:k}, \omega_k^i) - \nabla \frac{\delta F(\mu_{\mathscr{X}_k})}{\delta \mu}(X_k^i) \right\rangle \right] \mathrm{d}\widetilde{\mathscr{X}}_t$$

$$= \int \mathbb{E}_{\widetilde{\mathscr{X}}_0 | \widetilde{\mathscr{X}}_t} \left[ \left\langle \int \nabla_i \tilde{\mu}_t^{(N)}(\widetilde{\mathscr{X}}_t | \mathscr{X}_{0:k}') \mu^{(N)}(\mathrm{d}\mathscr{X}_{0:k}'), v_k^i((\widetilde{\mathscr{X}}_t, \mathscr{X}_{0:k-1}), \omega_k^i) - \nabla \frac{\delta F(\mu_{\widetilde{\mathscr{X}}_t})}{\delta \mu}(\tilde{X}_t^i) \right\rangle \right] \mathrm{d}\widetilde{\mathscr{X}}_t$$

$$+ \int \mathbb{E}_{\widetilde{\mathscr{X}}_0 | \widetilde{\mathscr{X}}_t} \left[ \left\langle \int \nabla_i \tilde{\mu}_t^{(N)}(\widetilde{\mathscr{X}}_t | \mathscr{X}_{0:k}') \mathrm{d}\mu^{(N)}(\mathscr{X}_{0:k}'), \right. \right.$$

$$\left. \left. v_k^i(\mathscr{X}_{0:k}, \omega_k^i) - \nabla \frac{\delta F(\mu_{\mathscr{X}_k})}{\delta \mu}(X_k^i) - v_k^i((\widetilde{\mathscr{X}}_t, \mathscr{X}_{0:k-1}), \omega_k^i) + \nabla \frac{\delta F(\mu_{\widetilde{\mathscr{X}}_t})}{\delta \mu}(\tilde{X}_t^i) \right\rangle \right] \mathrm{d}\widetilde{\mathscr{X}}_t, \tag{13}$$

where $(\widetilde{\mathscr{X}}_t, \mathscr{X}_{0:k-1}) = (\mathscr{X}_0, \mathscr{X}_1, \ldots, \mathscr{X}_{k-1}, \widetilde{\mathscr{X}}_t)$ which is obtained by replacing $\mathscr{X}_k$ of $\mathscr{X}_{0:k}$ to $\widetilde{\mathscr{X}}_t$.
First, we evaluate the first term in the right hand side of Eq. (13). Let

$$D_m^i F(\mathscr{X}) := \frac{\delta F(\mu_{\mathscr{X}})}{\delta \mu}(X^i),$$

for $\mathscr{X} = (X^i)_{i=1}^N$, and let $D_m F(\mathscr{X}) = (D_m^i F(\mathscr{X}))_{i=1}^N$. Let

$$\Delta v^i := v_k^i(\mathscr{X}_{0:k}', \omega_k^i) - D_m^i F(\mathscr{X}_k'),$$

and let $\Delta v = (\Delta v^i)_{i=1}^N$. We also define

$$Z_i = (\sqrt{2\lambda t})^{-1}[\widetilde{X}_t^i - (X_k'^i - tv_k^i(\mathscr{X}_{0:k}', \omega_k^i))].$$

Then $Z_i \sim N(0,1)$ conditioned by $\mathscr{X}_{0:k}'$ and is independent of $\omega_k^i$. We also define

$$v_k^i(\hat{\mathscr{X}}, \omega_k^i) := \mathbb{E}_{\mathscr{X}_{0:k-1}|\hat{\mathscr{X}}, \omega_{1:k}}[v_k^i((\hat{\mathscr{X}}, \mathscr{X}_{0:k-1}), \omega_k^i)],$$

where the expectation is taken conditioned by $\omega_{1:k}$ but we omit $\omega_{1:k-1}$ from the left hand side for the simplicity of notation. Since the density of the conditional distribution $\tilde{\mu}_t^{(N)}(\widetilde{\mathscr{X}}_t|\mathscr{X}_{0:k}')$ is proportional to $\exp\left(-\sum_{i=1}^N \frac{\|\widetilde{X}_t^i - (X_k'^i - tv_k^i(\mathscr{X}_{0:k}', \omega_k^i))\|^2}{2(2\lambda t)}\right)$, it holds that, under second order differentiability of $v_k^i$ and $D_m F$,

$$\int \left\langle \int -\frac{\widetilde{X}_t^i - (X_k'^i - tv_k^i(\mathscr{X}_{0:k}', \omega_k^i))}{2\lambda t} \tilde{\mu}_t^{(N)}(\widetilde{\mathscr{X}}_t|\mathscr{X}_{0:k}')\mu^{(N)}(d\mathscr{X}_{0:k}'),\right.$$

$$\left. v_k^i(\widetilde{\mathscr{X}}_t, \omega_k^i) - \nabla\frac{\delta F(\mu_{\widetilde{\mathscr{X}}_t})}{\delta\mu}(\widetilde{X}_t^i) \right\rangle d\widetilde{\mathscr{X}}_t$$

$$= \mathbb{E}_{Z, \mathscr{X}_{0:k}'}\left[\left\langle -\frac{Z_i}{\sqrt{2\lambda t}}, v_k^i(\mathscr{X}_k' - tD_m F(\mathscr{X}_k') + \sqrt{2\lambda t}Z - t\Delta v, \omega_k^i) - \right.\right.$$

$$\left.\left. D_m^i F(\mathscr{X}_k' - tD_m F(\mathscr{X}_k') + \sqrt{2\lambda t}Z - t\Delta v) \right\rangle \right]$$

$$= \mathbb{E}_{Z, \mathscr{X}_{0:k}'}\left[\left\langle -\frac{Z_i}{\sqrt{2\lambda t}}, \right.\right.$$

$$v_k^i(\mathscr{X}_k' - tD_m F(\mathscr{X}_k') + \sqrt{2\lambda t}Z, \omega_k^i) - D_m^i F(\mathscr{X}_k' - tD_m F(\mathscr{X}_k') + \sqrt{2\lambda t}Z)$$

$$+ v_k^i(\mathscr{X}_k' - tv_k(\mathscr{X}_{0:k}', \omega_k) + \sqrt{2\lambda t}Z, \omega_k^i) - v_k^i(\mathscr{X}_k' - tD_m F(\mathscr{X}_k') + \sqrt{2\lambda t}Z, \omega_k^i)$$

$$\left.\left. - D_m^i F(\mathscr{X}_k' - tv_k(\mathscr{X}_{0:k}', \omega_k) + \sqrt{2\lambda t}Z) + D_m^i F(\mathscr{X}_k' - tD_m F(\mathscr{X}_k') + \sqrt{2\lambda t}Z) \right\rangle \right]. \quad (14)$$

By Assumption 4, we can evaluate

$$\mathbb{E}_{\omega_k}\left[\left\|v_k^i(\mathscr{X}_k' - tv_k(\mathscr{X}_{0:k}', \omega_k) + \sqrt{2\lambda t}Z, \omega_k^i) - v_k^i(\mathscr{X}_k' - tD_m F(\mathscr{X}_k') + \sqrt{2\lambda t}Z, \omega_k^i)\right.\right.$$

$$\left.\left. - D_m^i F(\mathscr{X}_k' - tv_k(\mathscr{X}_{0:k}', \omega_k) + \sqrt{2\lambda t}Z) + D_m^i F(\mathscr{X}_k' - tD_m F(\mathscr{X}_k') + \sqrt{2\lambda t}Z)\right\|\right]$$

$$\leq \mathbb{E}_{\mathscr{X}_{0:k-1}'', \omega_k}\left[\left\|v_k^i((\mathscr{X}_k' - tv_k(\mathscr{X}_{0:k}', \omega_k) + \sqrt{2\lambda t}Z, \mathscr{X}_{0:k-1}''), \omega_k^i)\right.\right.$$

$$- v_k^i((\mathscr{X}_k' - tD_m F(\mathscr{X}_k') + \sqrt{2\lambda t}Z, \mathscr{X}_{0:k-1}''), \omega_k^i)$$

$$\left.\left. - D_m^i F(\mathscr{X}_k' - tv_k(\mathscr{X}_{0:k}', \omega_k) + \sqrt{2\lambda t}Z) + D_m^i F(\mathscr{X}_k' - tD_m F(\mathscr{X}_k') + \sqrt{2\lambda t}Z)\right\|\right]$$

$$\leq \mathbb{E}_{\mathscr{X}_{0:k-1}'', \omega_k}\left[\sum_{j=1}^N \left\|\nabla_j^\top v_k^i((\mathscr{X}_k' - tD_m F(\mathscr{X}_k') + \sqrt{2\lambda t}Z, \mathscr{X}_{0:k-1}''), \omega_k^i)\right.\right.$$

$$\left. - \nabla_j^\top D_m^i F((\mathscr{X}_k' - tD_m F(\mathscr{X}_k') + \sqrt{2\lambda t}Z, \mathscr{X}_{0:k-1}''))\right\|_{\text{op}} \|t\Delta v_j\|$$

$$\left. + Qt^2\left(\frac{1}{N}\sum_{j=1}^N \|\Delta v_j\|^2 + \|\Delta v_i\|^2\right)\right],$$

where Jensen's inequality is used in the first inequality. By using Assumption 4 again, the first term in the right hand side can be evaluated as

$$\sum_{j=1}^N \mathbb{E}_{\mathscr{X}_{0:k-1}'', \omega_k}\left[\frac{a_j}{2}\|\nabla_j^\top v_k^i((\mathscr{X}_k' - tD_m F(\mathscr{X}_k') + \sqrt{2\lambda t}Z, \mathscr{X}_{0:k-1}''), \omega_k^i)\right.$$

$$\left. - \nabla_j^\top D_m^i F((\mathscr{X}_k' - tD_m F(\mathscr{X}_k') + \sqrt{2\lambda t}Z, \mathscr{X}_{0:k-1}''))\|_{\text{op}}^2 + \frac{1}{2a_j}\|t\Delta v_j\|^2\right]$$

$$\leq (N-1)\left(\frac{a_{-i}}{2N^2}\tilde{\sigma}_{v,k}^2 + \frac{1}{2a_{-i}}t^2\sigma_{v,k}^2\right) + \left(\frac{a_i}{2}\tilde{\sigma}_{v,k}^2 + \frac{1}{2a_i}t^2\sigma_{v,k}^2\right),$$

for $a_j > 0$ $(j = 1, \ldots, N)$ where $a_j = a_{-i}$ $(j \neq i)$ for some $a_{-i} > 0$. By taking $a_{-i} = Nt\sigma_{v,k}/\tilde{\sigma}_{v,k}$ and $a_i = t\sigma_{v,k}/\tilde{\sigma}_{v,k}$, the right hand side can be bounded as

$$2t\tilde{\sigma}_{v,k}\sigma_{v,k}. \tag{15}$$

Then, we return to the right hand side of Eq. (14). First, we note that

$$\mathbb{E}_{\omega_k}\left[\left\langle -\frac{Z_i}{\sqrt{2\lambda t}}, v_k^i(\mathscr{X}_k' - tD_mF(\mathscr{X}_k') + \sqrt{2\lambda t}Z, \omega_k^i) - D_m^iF(\mathscr{X}_k' - tD_mF(\mathscr{X}_k') + \sqrt{2\lambda t}Z)\right\rangle\right] = 0,$$

for fixed $Z$ and $\mathscr{X}_k'$. Hence, by taking this and Eq. (15) into account, the expectation of the right hand side Eq. (14) with respect to $\omega_k$ can be further bounded as

$$\sqrt{2\frac{t}{\lambda}}\tilde{\sigma}_{v,k}\sigma_{v,k} + \sqrt{2}Qt^{3/2}\lambda^{-1/2}\sigma_{v,k}^2. \tag{16}$$

(2) Next, we evaluate the second term of the right hand side of Eq. (13). We have

$$\int \mathbb{E}_{\widetilde{\mathscr{X}_0}|\widetilde{\mathscr{X}_t}}\left[\left\langle \int \nabla_i\tilde{\mu}_t^{(N)}(\widetilde{\mathscr{X}_t}|\mathscr{X}_{0:k}')d\mu^{(N)}(\mathscr{X}_{0:k}'), \right.\right.$$
$$\left.\left. v_k^i(\mathscr{X}_k, \omega_k^i) - \nabla\frac{\delta F(\mu_{\mathscr{X}_k})}{\delta\mu}(X_k^i) - v_k^i(\widetilde{\mathscr{X}_t}, \omega_k^i) + \nabla\frac{\delta F(\mu_{\widetilde{\mathscr{X}_t}})}{\delta\mu}(\widetilde{X}_t^i)\right\rangle\right]d\widetilde{\mathscr{X}_t}$$

$$=\mathbb{E}_{\widetilde{\mathscr{X}_0},\widetilde{\mathscr{X}_t}}\left[\left\langle \nabla_i\log\left(\tilde{\mu}_t^{(N)}(\widetilde{\mathscr{X}_t})/p^{(N)}(\widetilde{\mathscr{X}_t})\right), \right.\right.$$
$$\left.\left. v_k^i(\mathscr{X}_k, \omega_k^i) - \nabla\frac{\delta F(\mu_{\mathscr{X}_k})}{\delta\mu}(X_k^i) - v_k^i(\widetilde{\mathscr{X}_t}, \omega_k^i) + \nabla\frac{\delta F(\mu_{\widetilde{\mathscr{X}_t}})}{\delta\mu}(\widetilde{X}_t^i)\right\rangle\right]$$

$$+\mathbb{E}_{\widetilde{\mathscr{X}_0},\widetilde{\mathscr{X}_t}}\left[\left\langle \nabla_i\log\left(p^{(N)}(\widetilde{\mathscr{X}_t})\right), \right.\right.$$
$$\left.\left. v_k^i(\mathscr{X}_k, \omega_k^i) - \nabla\frac{\delta F(\mu_{\mathscr{X}_k})}{\delta\mu}(X_k^i) - v_k^i(\widetilde{\mathscr{X}_t}, \omega_k^i) + \nabla\frac{\delta F(\mu_{\widetilde{\mathscr{X}_t}})}{\delta\mu}(\widetilde{X}_t^i)\right\rangle\right]. \tag{17}$$

The first term in the right hand side of this inequality (Eq. (17)) can be upper bounded by

$$\frac{\lambda}{4}\mathbb{E}\left[\left\|\nabla_i\log\left(\tilde{\mu}_t^{(N)}(\widetilde{\mathscr{X}_t})/p^{(N)}(\widetilde{\mathscr{X}_t})\right)\right\|^2\right]$$
$$+\frac{1}{\lambda}\int \mathbb{E}_{\widetilde{\mathscr{X}_0},\widetilde{\mathscr{X}_t}}\left[\left\|v_k^i(\mathscr{X}_k, \omega_k^i) - \nabla\frac{\delta F(\mu_{\mathscr{X}_k})}{\delta\mu}(X_k^i) - v_k^i(\widetilde{\mathscr{X}_t}, \omega_k^i) + \nabla\frac{\delta F(\mu_{\widetilde{\mathscr{X}_t}})}{\delta\mu}(\widetilde{X}_t^i)\right\|^2\right]$$

$$\leq\frac{\lambda}{4}\mathbb{E}\left[\left\|\nabla_i\log\left(\tilde{\mu}_t^{(N)}(\widetilde{\mathscr{X}_t})/p^{(N)}(\widetilde{\mathscr{X}_t})\right)\right\|^2\right]$$
$$+\frac{1}{\lambda}\int \mathbb{E}_{\widetilde{\mathscr{X}_0},\widetilde{\mathscr{X}_t}}\left[\left\|v_k^i((\mathscr{X}_k, \mathscr{X}_{0:k-1}), \omega_k^i) - \nabla\frac{\delta F(\mu_{\mathscr{X}_k})}{\delta\mu}(X_k^i) - v_k^i((\widetilde{\mathscr{X}_t}, \mathscr{X}_{0:k-1}), \omega_k^i) + \nabla\frac{\delta F(\mu_{\widetilde{\mathscr{X}_t}})}{\delta\mu}(\widetilde{X}_t^i)\right\|^2\right],$$

where we used Jensen's inequality. By Lemma 2 (the same bound applies to $v_k^i$ by the same proof), we see that the second term in the right hand side is bounded by

$$\frac{2}{\lambda}\delta_{\eta_k}. \tag{18}$$

Finally, we evaluate the second term in the right hand side of Eq. (17). Note that

$$\mathbb{E}_{\widetilde{\mathscr{X}_0},\widetilde{\mathscr{X}_t}}\left[\left\langle \nabla_i\log\left(p^{(N)}(\widetilde{\mathscr{X}_t})\right), \right.\right.$$

$$v_k^i(\mathscr{X}_k, \omega_k^i) - \nabla \frac{\delta F(\mu_{\mathscr{X}_k})}{\delta \mu}(X_k^i) - v_k^i(\widetilde{\mathscr{X}_t}, \omega_k^i) + \nabla \frac{\delta F(\mu_{\widetilde{\mathscr{X}_t}})}{\delta \mu}(\widetilde{X}_t^i) \Big\rangle \bigg]$$

$$= \mathbb{E}_{\widetilde{\mathscr{X}_0}, \widetilde{\mathscr{X}_t}} \left[ \Big\langle \nabla_i \log \left( p^{(N)}(\widetilde{\mathscr{X}_t}) \right) - \nabla_i \log \left( p^{(N)}(\widetilde{\mathscr{X}_0}) \right) + \nabla_i \log \left( p^{(N)}(\widetilde{\mathscr{X}_0}) \right), \right.$$

$$\left. v_k^i(\mathscr{X}_k, \omega_k^i) - \nabla \frac{\delta F(\mu_{\mathscr{X}_k})}{\delta \mu}(X_k^i) - v_k^i(\widetilde{\mathscr{X}_t}, \omega_k^i) + \nabla \frac{\delta F(\mu_{\widetilde{\mathscr{X}_t}})}{\delta \mu}(\widetilde{X}_t^i) \Big\rangle \right]$$

$$\leq \frac{2\delta_{\eta_k}}{\lambda} + \mathbb{E}_{\widetilde{\mathscr{X}_0}, \widetilde{\mathscr{X}_t}} \left[ \Big\langle \nabla_i \log \left( p^{(N)}(\widetilde{\mathscr{X}_0}) \right), \right.$$

$$\left. v_k^i(\mathscr{X}_k, \omega_k^i) - \nabla \frac{\delta F(\mu_{\mathscr{X}_k})}{\delta \mu}(X_k^i) - v_k^i(\widetilde{\mathscr{X}_t}, \omega_k^i) + \nabla \frac{\delta F(\mu_{\widetilde{\mathscr{X}_t}})}{\delta \mu}(\widetilde{X}_t^i) \Big\rangle \right],$$

where we used Lemma 2 with the same argument as Eq. (18) and Young's inequality. Taking the expectation with respect to $\omega_k^i$, it holds that

$$\mathbb{E}_{\omega_k^i} \left\{ \mathbb{E}_{\widetilde{\mathscr{X}_0}, \widetilde{\mathscr{X}_t}} \left[ \Big\langle \nabla_i \log \left( p^{(N)}(\widetilde{\mathscr{X}_0}) \right), v_k^i(\mathscr{X}_k, \omega_k^i) - \nabla \frac{\delta F(\mu_{\mathscr{X}_k})}{\delta \mu}(X_k^i) \Big\rangle \right] \right\} = 0.$$

Hence, it suffices to evaluate the term

$$- \mathbb{E}_{\widetilde{\mathscr{X}_0}, \widetilde{\mathscr{X}_t}} \left[ \Big\langle \nabla_i \log \left( p^{(N)}(\widetilde{\mathscr{X}_0}) \right), v_k^i(\widetilde{\mathscr{X}_t}, \omega_k^i) - \nabla \frac{\delta F(\mu_{\widetilde{\mathscr{X}_t}})}{\delta \mu}(\widetilde{X}_t^i) \Big\rangle \right].$$

Here, let $\hat{\mathscr{X}}_t = (\hat{X}_t^i)_{i=1}^n$ be the following stochastic process:

$$\hat{\mathscr{X}}_0 = \mathscr{X}_k,$$
$$\mathrm{d}\hat{X}_t^i = -\nabla \frac{\delta F(\mu_k)}{\delta \mu}(X_k^i)\mathrm{d}t + \sqrt{2\lambda}\mathrm{d}W_t^i,$$

for $0 \leq t \leq \eta$, where $(W_t^i)_t$ is the same Brownian motion as that drives $\widetilde{X}_t^i$. Then, the term we are interested in can be evaluated as

$$- \mathbb{E}_{\widetilde{\mathscr{X}_0}, \widetilde{\mathscr{X}_t}, \omega_k} \left[ \Big\langle \nabla_i \log \left( p^{(N)}(\widetilde{\mathscr{X}_0}) \right), v_k^i(\widetilde{\mathscr{X}_t}, \omega_k^i) - \nabla \frac{\delta F(\mu_{\widetilde{\mathscr{X}_t}})}{\delta \mu}(\widetilde{X}_t^i) \Big\rangle \right]$$

$$= - \mathbb{E}_{\widetilde{\mathscr{X}_0}, \widetilde{\mathscr{X}_t}, \hat{\mathscr{X}}_t, \omega_k} \left[ \Big\langle \nabla_i \log \left( p^{(N)}(\widetilde{\mathscr{X}_0}) \right), v_k^i(\widetilde{\mathscr{X}_t}, \omega_k^i) - \nabla \frac{\delta F(\mu_{\widetilde{\mathscr{X}_t}})}{\delta \mu}(\widetilde{X}_t^i) \right.$$

$$\left. - v_k^i(\hat{\mathscr{X}}_t, \omega_k^i) - \nabla \frac{\delta F(\mu_{\hat{\mathscr{X}}_t})}{\delta \mu}(\hat{X}_t^i) + v_k^i(\hat{\mathscr{X}}_t, \omega_k^i) - \nabla \frac{\delta F(\mu_{\hat{\mathscr{X}}_t})}{\delta \mu}(\hat{X}_t^i) \Big\rangle \right]$$

$$= - \mathbb{E}_{\widetilde{\mathscr{X}_0}, \widetilde{\mathscr{X}_t}, \hat{\mathscr{X}}_t, \omega_k} \left[ \Big\langle \nabla_i \log \left( p^{(N)}(\widetilde{\mathscr{X}_0}) \right), v_k^i(\widetilde{\mathscr{X}_t}, \omega_k^i) - \nabla \frac{\delta F(\mu_{\widetilde{\mathscr{X}_t}})}{\delta \mu}(\widetilde{X}_t^i) \right.$$

$$\left. - v_k^i(\hat{\mathscr{X}}_t, \omega_k^i) - \nabla \frac{\delta F(\mu_{\hat{\mathscr{X}}_t})}{\delta \mu}(\hat{X}_t^i) \Big\rangle \right]$$

$$\leq \mathbb{E}_{\widetilde{\mathscr{X}_0}, \widetilde{\mathscr{X}_t}, \hat{\mathscr{X}}_t} \left\{ \left\| \nabla_i \log \left( p^{(N)}(\widetilde{\mathscr{X}_0}) \right) \right\| \cdot \right.$$

$$\left. \mathbb{E}_{\omega_k} \left[ \left\| v_k^i(\widetilde{\mathscr{X}_t}, \omega_k^i) - \nabla \frac{\delta F(\mu_{\widetilde{\mathscr{X}_t}})}{\delta \mu}(\widetilde{X}_t^i) - v_k^i(\hat{\mathscr{X}}_t, \omega_k^i) - \nabla \frac{\delta F(\mu_{\hat{\mathscr{X}}_t})}{\delta \mu}(\hat{X}_t^i) \right\| \right] \right\}. \qquad (19)$$

With the same reasoning as in Eq. (16), it holds that

$$\mathbb{E}_{\omega_k} \left[ \left\| v_k^i(\widetilde{\mathscr{X}_t}, \omega_k^i) - \nabla \frac{\delta F(\mu_{\widetilde{\mathscr{X}_t}})}{\delta \mu}(\hat{X}_t^i) - v_k^i(\hat{\mathscr{X}}_t, \omega_k^i) - \nabla \frac{\delta F(\mu_{\hat{\mathscr{X}}_t})}{\delta \mu}(\widetilde{X}_t^i) \right\| \right]$$

$$\leq 2t\tilde{\sigma}_{v,k}\sigma_{v,k} + 2Qt^2\sigma_{v,k}^2.$$

Moreover, Assumption 2 and Lemm 1 yield that

$$\mathbb{E}\left[\left\|\nabla_i \log\left(p^{(N)}(\widetilde{\mathscr{X}_0})\right)\right\|\right] \leq \frac{1}{\lambda}\left(\sup_{\mu \in \mathcal{P}, x \in \mathbb{R}^d}\left\|\nabla\frac{\delta U(\mu)}{\delta\mu}(x)\right\| + \mathbb{E}[\|\nabla r(\widetilde{X}_0^i)\|]\right)$$

$$\leq \frac{1}{\lambda}\left(R + \mathbb{E}[\|\nabla r(\widetilde{X}_0^i) - r(0) + r(0)\|]\right)$$

$$\leq \frac{1}{\lambda}\left(R + \lambda_2\mathbb{E}[\|\widetilde{X}_0^i\|]\right)$$

$$\leq \frac{1}{\lambda}\left(R + \lambda_2\sqrt{\mathbb{E}[\|\widetilde{X}_0^i\|^2]}\right)$$

$$\leq \frac{1}{\lambda}\left(R + \lambda_2\bar{R}\right),$$

where the second and third inequalities are due to Assumption 2 and the last inequality is by Lemma 1. Then, the right hand side of Eq. (19) can be bounded by

$$\frac{1}{\lambda}\left(R + \lambda_2\bar{R}\right)\left(2t\tilde{\sigma}_{v,k}\sigma_{v,k} + 2Qt^2\sigma_{v,k}^2\right).$$

where we used Assumption 2 and Lemm 1. Hence, the second term of the right hand side of Eq. (13) (which is same as Eq. (17)) can be bounded by

$$\frac{4}{\lambda}\delta_{\eta_k} + \frac{1}{\lambda}\left(R + \lambda_2\bar{R}\right)\left(2t\tilde{\sigma}_{v,k}\sigma_{v,k} + 2Qt^2\sigma_{v,k}^2\right).$$

(3) By summing up the results in (1) and (2), we have that

$$C\text{-(I)} \leq \frac{\lambda}{4}\mathbb{E}\left[\left\|\nabla_i \log\left(\tilde{\mu}_t^{(N)}(\widetilde{\mathscr{X}_t})/p^{(N)}(\widetilde{\mathscr{X}_t})\right)\right\|^2\right]$$

$$+ 4\delta_{\eta_k} + \left(R + \lambda_2\bar{R}\right)\left(1 + \sqrt{\frac{\lambda}{t}}\right)\left(2\frac{t}{\lambda}\tilde{\sigma}_{v,k}\sigma_{v,k} + 2\frac{Qt^2}{\lambda}\sigma_{v,k}^2\right).$$

*Evaluation of $C$-(II):*

Next, we evaluate the term $C$-(II). Here, let $\hat{\mathscr{X}_t} = (\hat{X}_t^i)_{i=1}^n$ be the following stochastic process:

$$\hat{\mathscr{X}}_0 = \mathscr{X}_k,$$

$$\mathrm{d}\hat{X}_t^i = -\nabla\frac{\delta F(\mu_k)}{\delta\mu}(X_k^i)\mathrm{d}t + \sqrt{2\lambda}\mathrm{d}W_t^i,$$

for $0 \leq t \leq \eta$, where $(W_t^i)_t$ is the same Brownian motion as that drives $\widetilde{X}_t^i$. By definition, we notice that

$$\mathrm{d}(\widetilde{X}_t^i - \hat{X}_t^i) = \left(\nabla\frac{\delta F(\mu_k)}{\delta\mu}(X_k^i) - v_k^i\right)\mathrm{d}t,$$

which yields that

$$\widetilde{X}_t^i - \hat{X}_t^i = t\left(\nabla\frac{\delta F(\mu_k)}{\delta\mu}(X_k^i) - v_k^i\right). \tag{20}$$

By subtracting and adding the derivative at $\hat{\mathscr{X}_t}$ from $\nabla_i \log(p^{(N)}(\widetilde{\mathscr{X}_t}))$, we have

$$\nabla_i \log\left(p^{(N)}(\widetilde{\mathscr{X}_t})\right) = -\frac{1}{\lambda}\nabla\frac{\delta F(\mu_{\widetilde{\mathscr{X}_t}})}{\delta\mu}(\widetilde{X}_t^i)$$

$$= -\frac{1}{\lambda}\left(\nabla\frac{\delta F(\mu_{\widetilde{\mathscr{X}_t}})}{\delta\mu}(\widetilde{X}_t^i) - \nabla\frac{\delta F(\mu_{\hat{\mathscr{X}_t}})}{\delta\mu}(\hat{X}_t^i)\right) - \frac{1}{\lambda}\nabla\frac{\delta F(\mu_{\hat{\mathscr{X}_t}})}{\delta\mu}(\hat{X}_t^i).$$

By Assumptions 1 and 2, the first two terms in the right hand side can be bounded as

$$\left\|\nabla\frac{\delta F(\mu_{\widetilde{\mathscr{X}_t}})}{\delta\mu}(\widetilde{X}_t^i) - \nabla\frac{\delta F(\mu_{\hat{\mathscr{X}_t}})}{\delta\mu}(\hat{X}_t^i)\right\|$$

$$= \left\| \nabla \frac{\delta U(\mu_{\widetilde{\mathscr{X}_t}})}{\delta \mu}(\widetilde{X}_t^i) + \nabla r(\widetilde{X}_t^i) - \nabla \frac{\delta U(\mu_{\hat{\mathscr{X}_t}})}{\delta \mu}(\hat{X}_t^i) - \nabla r(\hat{X}_t^i) \right\|$$

$$\leq L(W_2(\mu_{\widetilde{\mathscr{X}_t}}, \mu_{\hat{\mathscr{X}_t}}) + \|\widetilde{X}_t^i - \hat{X}_t^i\|) + \lambda_2 \|\widetilde{X}_t^i - \hat{X}_t^i\|$$

$$\leq t(L + \lambda_2) \left( \sqrt{\frac{1}{N} \sum_{i=1}^N \left\| v_k^i - \nabla \frac{\delta F(\mu_k)}{\delta \mu}(X_k^i) \right\|^2} + \left\| v_k^i - \nabla \frac{\delta F(\mu_k)}{\delta \mu}(X_k^i) \right\| \right), \qquad (21)$$

where the first inequality follows from

$$\|\nabla r(x) - \nabla r(x')\| = \left\| \int_0^1 [\nabla \nabla^\top r(\theta x + (1-t)x)](x - x') \mathrm{d}\theta \right\|$$

$$\leq \lambda_2 \int_0^1 \|x - x'\| \mathrm{d}\theta = \lambda_2 \|x - x'\| \qquad (22)$$

by Assumption 1 and we used Eq. (20) in the last inequality. Therefore, by noticing $\widetilde{\mathscr{X}_0} = \mathscr{X}_k$, we can obtain the following evaluation:

$$\sum_{i=1}^N C\text{-(II)}$$

$$= \sum_{i=1}^N \left| \mathbb{E}_{v_k^i} \left[ \mathbb{E}_{\widetilde{\mathscr{X}_t}, \mathscr{X}_{0:k}} \left[ \left\langle \nabla_i \log \left( p^{(N)}(\widetilde{\mathscr{X}_t}) \right), v_k^i(\mathscr{X}_{0:k}) - \nabla \frac{\delta F(\mu_{\widetilde{\mathscr{X}_0}})}{\delta \mu}(\widetilde{X}_0^i) \right\rangle \right] \right] \right|$$

$$\leq \frac{2(L + \lambda_2)^2}{\lambda} t \sum_{i=1}^N \mathbb{E}_{(v_k^i)_{i=1}^N, \mathscr{X}_{0:k}} \left[ \left\| v_k^i(\mathscr{X}_{0:k}) - \nabla \frac{\delta F(\mu_{\widetilde{\mathscr{X}_0}})}{\delta \mu}(\widetilde{X}_0^i) \right\|^2 \right]$$

$$\leq t N \frac{2(L + \lambda_2)^2}{\lambda} \sigma_{v,k}^2.$$

*Combining the bounds on $C$-(I) and $C$-(II):*

$$\sum_{i=1}^N \lambda C \leq \lambda \sum_{i=1}^N (C\text{-(I)} + C\text{-(II)})$$

$$\leq \frac{\lambda^2}{4} \sum_{i=1}^N \mathbb{E} \left[ \left\| \nabla_i \log \left( \tilde{\mu}_t^{(N)}(\widetilde{\mathscr{X}_t})/p^{(N)}(\widetilde{\mathscr{X}_t}) \right) \right\|^2 \right]$$

$$+ N \left\{ 4\delta_{\eta_k} + \left( R + \lambda_2 \bar{R} \right) \left( 1 + \sqrt{\frac{\lambda}{t}} \right) (2t\tilde{\sigma}_{v,k}\sigma_{v,k} + 2Qt^2\sigma_{v,k}^2) + t[2(L + \lambda_2)^2\sigma_{v,k}^2] \right\}.$$

## B.3 Putting Things Together

By Gronwall lemma (see Mischler (2019), for example), we arrive at

$$\frac{1}{N} \mathbb{E}_{\omega_{0:k}}[\mathscr{F}^N(\mu_{k+1}^{(N)})] - \mathscr{F}(\mu^*)$$

$$\leq \exp(-\lambda \alpha \eta_k/2) \left( \frac{1}{N} \mathbb{E}_{\omega_{0:k-1}}[\mathscr{F}^N(\mu_k^{(N)})] - \mathscr{F}(\mu^*) \right) + \eta_k \left( \delta_{\eta_k} + \frac{C_\lambda}{N} \right) + \sigma_{v,k}^2 \eta_k.$$

In addition, if Assumption 4 is satisfied, we have

$$\frac{1}{N} \mathbb{E}_{\omega_{0:k}}[\mathscr{F}^N(\mu_{k+1}^{(N)})] - \mathscr{F}(\mu^*)$$

$$\leq \exp(-\lambda \alpha \eta_k/2) \left( \frac{1}{N} \mathbb{E}_{\omega_{0:k-1}}[\mathscr{F}^N(\mu_k^{(N)})] - \mathscr{F}(\mu^*) \right) + \eta_k \left( \delta_{\eta_k} + \frac{C_\lambda}{N} \right)$$

$$+ \left[ 4\eta_k \delta_{\eta_k} + \left( R + \lambda_2 \bar{R} \right) \left( 1 + \sqrt{\frac{\lambda}{\eta_k}} \right) (\eta_k^2 \tilde{\sigma}_{v,k}\sigma_{v,k} + Q\eta_k^3\sigma_{v,k}^2) + \eta_k^2(L + \lambda_2)^2\sigma_{v,k}^2 \right].$$

# C  Proof of Theorem 3: F-MFLD and SGD-MFLD

Applying the Gronwall lemma (Mischler, 2019) with Theorem 2, we have that

$$\frac{1}{N}\mathbb{E}_{\omega_{0:k-1}}[\mathscr{F}^N(\mu_k^{(N)})] - \mathscr{F}(\mu^*)$$

$$\leq \exp\left(-\lambda\alpha\sum_{j=0}^{k-1}\eta_j/2\right)\Delta_0 + \sum_{i=0}^{k-1}\exp\left(-\lambda\alpha\sum_{j=i}^{k-1}\eta_j/2\right)\left[\Upsilon_i + \eta_i\left(\delta_{\eta_i} + \frac{C_\lambda}{N}\right)\right]. \quad (23)$$

Here, remember that Assumption 5 yields that

$$\sigma_{v,k}^2 \leq \frac{R^2}{B}, \quad \tilde{\sigma}_{v,k}^2 \leq \frac{R^2}{B}.$$

Under Assumption 5, we can easily check that Assumption 4 holds with $Q = R$. When $\eta_k = \eta$ for all $k \geq 0$, we have a uniform upper bound of $\Upsilon_k$ as

$$\Upsilon_k \leq \begin{cases} 4\eta\delta_\eta + \left[R + \lambda_2\bar{R} + (L+\lambda_2)^2\right]\left(1 + \sqrt{\frac{\lambda}{\eta}}\right)\eta^2\frac{R^2}{B} \\ \quad + \left(R + \lambda_2\bar{R}\right)R\left(1 + \sqrt{\frac{\lambda}{\eta}}\right)\eta^3\frac{R^2}{B}, & \text{with Assumption 5-(ii),} \\ \frac{R^2}{B}\eta, & \text{without Assumption 5-(ii).} \end{cases}$$

We denote the right hand side as $\bar{\Upsilon}$. Then we have

$$\frac{1}{N}\mathbb{E}_{\omega_{0:k-1}}[\mathscr{F}^N(\mu_k^{(N)})] - \mathscr{F}(\mu^*)$$

$$\leq \exp\left(-\lambda\alpha\eta k/2\right)\Delta_0 + \sum_{i=0}^{k-1}\exp\left(-\lambda\alpha(k-1-i)\eta/2\right)\left[\bar{\Upsilon} + \eta\left(\delta_\eta + \frac{C_\lambda}{N}\right)\right]$$

$$\leq \exp\left(-\lambda\alpha\eta k/2\right)\Delta_0 + \frac{1 - \exp(-\lambda\alpha k\eta/2)}{1 - \exp(-\lambda\alpha\eta/2)}\left[\bar{\Upsilon} + \eta\left(\delta_\eta + \frac{C_\lambda}{N}\right)\right]$$

$$\leq \exp\left(-\lambda\alpha\eta k/2\right)\Delta_0 + \frac{4}{\lambda\alpha\eta}\left[\bar{\Upsilon} + \eta\left(\delta_\eta + \frac{C_\lambda}{N}\right)\right] \quad (\because \lambda\alpha\eta/2 \leq 1/2)$$

$$= \exp\left(-\lambda\alpha\eta k/2\right)\Delta_0 + \frac{4}{\lambda\alpha}\bar{L}^2 C_1\left(\lambda\eta + \eta^2\right) + \frac{4}{\lambda\alpha\eta}\bar{\Upsilon} + \frac{4C_\lambda}{\lambda\alpha N}.$$

Then, by taking

$$k \geq \frac{2}{\lambda\alpha\eta}\log(\Delta_0/\epsilon),$$

then the right hand side can be bounded as

$$\frac{1}{N}\mathbb{E}[\mathscr{F}^N(\mu_k^{(N)})] - \mathscr{F}(\mu^*)$$

$$\leq \epsilon + \frac{4}{\lambda\alpha}\bar{L}^2 C_1\left(\lambda\eta + \eta^2\right) + \frac{4}{\lambda\alpha\eta}\bar{\Upsilon} + \frac{4C_\lambda}{\lambda\alpha N}.$$

(1) Hence, without Assumption 5-(ii), if we take

$$\eta \leq \frac{\lambda\alpha\epsilon}{8\bar{L}^2}\left(C_1\lambda\right)^{-1} \wedge \frac{1}{8\bar{L}}\sqrt{\frac{2\lambda\alpha\epsilon}{C_1}}, \quad B \geq 4R^2/(\lambda\alpha\epsilon)$$

then the right hand side can be bounded as

$$\frac{1}{N}\mathbb{E}[\mathscr{F}^N(\mu_k^{(N)})] - \mathscr{F}(\mu^*) \leq 3\epsilon + \frac{4C_\lambda}{\lambda\alpha N}.$$

We can easily check that the number of iteration $k$ satisfies

$$k = O\left(\frac{\bar{L}^2}{\lambda\alpha\epsilon}\lambda + \frac{\bar{L}}{\sqrt{\lambda\alpha\epsilon}}\right)\frac{1}{\lambda\alpha}\log(\epsilon^{-1}),$$

in this setting.

(2) On the other hand, under Assumption 5-(ii), if we take

$$\eta \leq \frac{\lambda\alpha\epsilon}{40\bar{L}^2}\left(C_1\lambda\right)^{-1} \wedge \frac{1}{\bar{L}}\sqrt{\frac{\lambda\alpha\epsilon}{40C_1}} \wedge 1,$$

$$B \geq 4\left[(1+R)(R+\lambda_2\bar{R})+(L+\lambda_2)^2\right]R^2(\eta+\sqrt{\eta\lambda})/(\lambda\alpha\epsilon),$$

then the right hand side can be bounded as

$$\frac{1}{N}\mathbb{E}[\mathscr{F}^N(\mu_k^{(N)})] - \mathscr{F}(\mu^*) \leq 3\epsilon + \frac{4C_\lambda}{\lambda\alpha N}.$$

We can again check that it suffices to take the number of iteration $k$ as

$$k = O\left(\frac{\bar{L}^2}{\lambda\alpha\epsilon}\lambda + \frac{\bar{L}}{\sqrt{\lambda\alpha\epsilon}}\right)\frac{1}{\lambda\alpha}\log(\epsilon^{-1}),$$

to achieve the targe accuracy.

## D Proof of Theorem 4: SVRG-MFLD

The standard argument of variance yields

$$\sigma_{v,k}^2 \leq \max_{1\leq i\leq N}\frac{n-B}{B(n-1)}\frac{1}{n}\sum_{j=1}^{n}\mathbb{E}[\|\Delta_k^{i,j}\|^2],$$

where $\Delta_k^{i,j} = \nabla\frac{\delta\ell_j(\mu_k^{(N)})}{\delta\mu}(X_k^i) - \nabla\frac{\delta U(\mu_k^{(N)})}{\delta\mu}(X_k^i) - \nabla\frac{\delta\ell_i(\mu_s^{(N)})}{\delta\mu}(X_s^i) + \nabla\frac{\delta U(\mu_s^{(N)})}{\delta\mu}(X_s^i)$. Here, Assumption 2 and 6 give that

$$\|\Delta_k^{i,j}\|^2 \leq 4(L^2W_2^2(\mu_k^{(N)},\mu_s^{(N)})+L^2\|X_k^i-X_s^i\|^2) \leq 4L^2\left(\frac{1}{N}\sum_{j'=1}^{N}\|X_k^{j'}-X_s^{j'}\|^2+\|X_k^i-X_s^i\|^2\right).$$

Taking the expectation, we have

$$\frac{1}{n}\sum_{j=1}^{n}\mathbb{E}[\|\Delta_k^{i,j}\|^2] \leq 4L^2\frac{1}{n}\sum_{j=1}^{n}\sum_{l=s}^{k-1}(\eta^2\mathbb{E}[\|v_l^i\|^2]+2\eta\lambda d)$$

$$\leq C_1L^2\underbrace{(k-s)}_{\leq m}(\eta^2+\eta\lambda),$$

where we used that $\mathbb{E}[\|v_k^i\|^2] \leq 2(R^2+\lambda_2(c_r+\bar{R}^2)) \leq C_1$ by Eq. (24) with Lemma 1. Hence, we have that

$$\sigma_{v,k}^2 \leq C_1\Xi L^2m(\eta^2+\eta\lambda),$$

where $\Xi = \frac{n-B}{B(n-1)}$.

On the other hand, we have

$$\tilde{\sigma}_{v,k}^2 \leq \frac{n-B}{B(n-1)}R^2 = \Xi R^2,$$

by the same argument with SGD-MFLD. We have a uniform upper bound of $\Upsilon_k$ as

$$\Upsilon_k \leq \begin{cases} 4\eta\delta_\eta + \left(R+\lambda_2\bar{R}\right)\left(1+\sqrt{\frac{\lambda}{\eta}}\right)\eta^2\sqrt{C_1\Xi L^2m(\eta^2+\eta\lambda)}\sqrt{R^2\Xi} \\ \quad + \left[(R+\lambda_2\bar{R})R\eta^3+(L+\lambda_2)^2\eta^2\right]\left(1+\sqrt{\frac{\lambda}{\eta}}\right)C_1\Xi L^2m(\eta^2+\eta\lambda), & \text{with Assumption 6-(ii),} \\ \eta C_1\Xi L^2m(\eta^2+\eta\lambda), & \text{without Assumption 6-(ii).} \end{cases}$$

We again let the right hand side be $\bar{\Upsilon}$. Then, we have that

$$\frac{1}{N}\mathbb{E}[\mathscr{F}^N(\mu_k^{(N)})] - \mathscr{F}(\mu^*)$$

$$\leq \exp(-\lambda\alpha\eta/2)\left(\frac{1}{N}\mathbb{E}[\mathscr{F}^N(\mu_{k-1}^{(N)})] - \mathscr{F}(\mu^*)\right) + \bar{\Upsilon} + \eta\left(\delta_\eta + \frac{C_\lambda}{N}\right).$$

Then, by the Gronwall's lemma yields that

$$\frac{1}{N}\mathbb{E}[\mathscr{F}^N(\mu_k^{(N)})] - \mathscr{F}(\mu^*)$$

$$\leq \exp(-\lambda\alpha\eta k/2)\left(\frac{1}{N}\mathbb{E}[\mathscr{F}^N(\mu_0^{(N)})] - \mathscr{F}(\mu^*)\right) + \sum_{l=1}^{k}\exp(-\lambda\alpha\eta(k-l)/2)\left[\bar{\Upsilon} + \eta\left(\delta_\eta + \frac{C_\lambda}{N}\right)\right]$$

$$\leq \exp(-\lambda\alpha\eta k/2)\left(\frac{1}{N}\mathbb{E}[\mathscr{F}^N(\mu_0^{(N)})] - \mathscr{F}(\mu^*)\right) + \frac{1-\exp(-\lambda\alpha\eta k/2)}{1-\exp(-\lambda\alpha\eta/2)}\left[\bar{\Upsilon} + \eta\left(\delta_\eta + \frac{C_\lambda}{N}\right)\right]$$

$$\leq \exp(-\lambda\alpha\eta k/2)\left(\frac{1}{N}\mathbb{E}[\mathscr{F}^N(\mu_0^{(N)})] - \mathscr{F}(\mu^*)\right) + \frac{4}{\lambda\alpha\eta}\bar{\Upsilon} + \frac{4}{\alpha\lambda}\left(\delta_\eta + \frac{C_\lambda}{N}\right),$$

where we used $\lambda\alpha\eta/2 \leq 1/2$ in the last inequality.

(1) Without Assumption 6-(ii), we have

$$\frac{1}{\lambda\alpha\eta}\bar{\Upsilon} = \frac{C_1}{\lambda\alpha\eta}\Xi L^2 m\eta^2(\eta+\lambda) = \frac{C_1}{\lambda\alpha}L^2 m\eta(\eta+\lambda)\frac{(n-B)}{B(n-1)},$$

we obtain that

$$\frac{1}{N}\mathbb{E}[\mathscr{F}^N(\mu_k^{(N)})] - \mathscr{F}(\mu^*) \leq \epsilon + \frac{4C_1\bar{L}^2(\eta^2+\lambda\eta) + 4C_\lambda/N}{\lambda\alpha} + \frac{4C_1}{\lambda\alpha}L^2 m\eta(\eta+\lambda)\frac{(n-B)}{B(n-1)}$$

$$= \epsilon + \frac{4C_1\bar{L}^2(\eta^2+\lambda\eta)(1+\frac{(n-B)}{B(n-1)}m)}{\lambda\alpha} + \frac{4C_\lambda}{\lambda\alpha N}$$

when

$$k \gtrsim \frac{1}{\lambda\alpha\eta}\log(\epsilon^{-1}),$$

and $\lambda\alpha\eta/2 \leq 1/2$. In particular, if we set $B \geq m$ and

$$\eta = \frac{\alpha\epsilon}{4\bar{L}^2 C_1} \wedge \frac{\sqrt{\lambda\alpha\epsilon}}{4\bar{L}\sqrt{C_1}},$$

we have

$$\frac{1}{N}\mathbb{E}[\mathscr{F}^N(\mu_k^{(N)})] - \mathscr{F}(\mu^*) \leq \epsilon + \frac{4C_\lambda}{\alpha\lambda N},$$

with the iteration complexity and the total gradient computation complexity:

$$k \lesssim \left(\frac{\bar{L}^2}{\alpha\epsilon} + \frac{\bar{L}}{\sqrt{\lambda\alpha\epsilon}}\right)\frac{1}{(\lambda\alpha)}\log(\epsilon^{-1}),$$

$$Bk + \frac{nk}{m} + n \lesssim \sqrt{n}k + n \lesssim \sqrt{n}\left(\frac{\bar{L}^2}{\alpha\epsilon} + \frac{\bar{L}}{\sqrt{\lambda\alpha\epsilon}}\right)\frac{1}{(\lambda\alpha)}\log(\epsilon^{-1}) + n,$$

where $m = B = \sqrt{n}$.

(2) With Assumption 6-(ii), we have

$$\frac{1}{\lambda\alpha\eta}\bar{\Upsilon} = \frac{4}{\lambda\alpha}\delta_\eta + \frac{1}{\lambda\alpha}O\left(\eta\sqrt{m(\eta^2+\lambda\eta)} + \eta m(\eta^2+\lambda\eta)\right)\left(1+\sqrt{\frac{\lambda}{\eta}}\right)\frac{(n-B)}{B(n-1)}.$$

Then, we obtain that

$$\frac{1}{N}\mathbb{E}[\mathscr{F}^N(\mu_k^{(N)})] - \mathscr{F}(\mu^*)$$

$$\leq \epsilon + \frac{20C_1\bar{L}^2(\eta^2+\lambda\eta) + 4C_\lambda/N}{\lambda\alpha} + \frac{1}{\lambda\alpha}O\left(\eta\sqrt{m(\eta^2+\lambda\eta)} + \eta m(\eta^2+\lambda\eta)\right)\left(1+\sqrt{\frac{\lambda}{\eta}}\right)\frac{(n-B)}{B(n-1)},$$

when
$$k \gtrsim \frac{1}{\lambda \alpha \eta} \log(\epsilon^{-1}),$$
and $\lambda \alpha \eta / 2 \leq 1/2$. In particular, if we set $B \geq \left[ \sqrt{m} \frac{(\eta + \sqrt{\eta/\lambda})^2}{\eta \lambda} \vee m \frac{(\eta + \sqrt{\eta/\lambda})^3}{\eta \lambda} \right] \wedge n$ and
$$\eta = \frac{\alpha \epsilon}{40 \bar{L}^2 C_1} \wedge \frac{\sqrt{\lambda \alpha \epsilon}}{\bar{L}\sqrt{40 C_1}},$$
we have
$$\frac{1}{N} \mathbb{E}[\mathscr{F}^N(\mu_k^{(N)})] - \mathscr{F}(\mu^*) \leq O(\epsilon) + \frac{4 C_\lambda}{\alpha \lambda N},$$
with the iteration complexity and the total gradient computation complexity:
$$k \lesssim \left( \frac{\bar{L}^2}{\alpha \epsilon} + \frac{\bar{L}}{\sqrt{\lambda \alpha \epsilon}} \right) \frac{1}{(\lambda \alpha)} \log(\epsilon^{-1}),$$
$$Bk + \frac{nk}{m} + n \lesssim \max\left\{ n^{1/3} \left(1 + \sqrt{\eta/\lambda}\right)^{4/3}, \sqrt{n} \left(1 + \sqrt{\eta/\lambda}\right)^{3/2} (\sqrt{\eta \lambda})^{1/2} \right\} k + n$$
$$\lesssim \max\left\{ n^{1/3} \left(1 + \sqrt{\eta/\lambda}\right)^{4/3}, \sqrt{n} \left(1 + \sqrt{\eta/\lambda}\right)^{3/2} (\sqrt{\eta \lambda})^{1/2} \right\} \eta^{-1} \frac{1}{(\lambda \alpha)} \log(\epsilon^{-1}) + n,$$
where $m = \Omega(n/B) = \Omega([n^{2/3}(1 + \sqrt{\eta/\lambda})^{-4/3} \wedge \sqrt{n}(1 + \sqrt{\eta/\lambda})^{-3/2}(\sqrt{\eta \lambda})^{-1/2})] \vee 1)$ and $B \geq \left[ n^{\frac{1}{3}} \left(1 + \sqrt{\frac{\eta}{\lambda}}\right)^{\frac{4}{3}} \vee \sqrt{n}(\eta \lambda)^{\frac{1}{4}} \left(1 + \sqrt{\frac{\eta}{\lambda}}\right)^{\frac{3}{2}} \right] \wedge n.$

# E  Auxiliary lemmas

## E.1  Properties of the MFLD Iterates

Under Assumption 2 with Assumption 5 for SGD-MFLD or 6 for SVRG-MFLD, we can easily verify that
$$\left\| \nabla \frac{\delta U(\mu_k)}{\delta \mu}(X_k^i) \right\| \leq R, \quad \mathbb{E}_{\omega_k^i | \mathscr{X}_{0:k}}[\|v_k^i\|^2] \leq 2(R^2 + \lambda_2(c_r + \|X_k^i\|^2)). \tag{24}$$

**Lemma 1.** *Under Assumption 2 with Assumption 5 for SGD-MFLD or Assumption 6 for SVRG-MFLD, if $\eta \leq \lambda_1/(4\lambda_2)$, we have the following uniform bound of the second moment of $X_k^i$:*
$$\mathbb{E}[\|X_k^i\|^2] \leq \mathbb{E}[\|X_0^i\|^2] + \frac{2}{\lambda_1} \left[ \left( \frac{\lambda_1}{8\lambda_2} + \frac{1}{2\lambda_1} \right) (R^2 + \lambda_2 c_r) + \lambda d \right],$$
*for any $k \geq 1$.*

*Proof.* By the update rule of $X_k^i$ and Assumption 1, we have
$$\mathbb{E}[\|X_{k+1}^i\|^2] = \mathbb{E}[\|X_k^i\|^2] - 2\mathbb{E}[\langle X_k^i, \eta v_k^i + \sqrt{2\eta \lambda} \xi_k^i \rangle] + \mathbb{E}[\|\eta v_k^i + \sqrt{2\eta \lambda} \xi_k^i\|^2]$$
$$= \mathbb{E}[\|X_k^i\|^2] - 2\eta \mathbb{E}\left[ \left\langle X_k^i, \nabla \frac{\delta U(\mu_k)}{\delta \mu}(X_k^i) + \nabla r(X_k^i) \right\rangle \right] + \mathbb{E}[\eta^2 \|v_k^i\|^2] + 2\eta \lambda d$$
$$\leq \mathbb{E}[\|X_k^i\|^2] + 2\eta R \mathbb{E}[\|X_k^i\|] - 2\eta \lambda_1 \mathbb{E}[\|X_k^i\|^2] + 2\eta^2(R^2 + \lambda_2(c_r + \|X_k^i\|^2)) + 2\eta \lambda d$$
$$\leq (1 - 2\eta \lambda_1 + 2\eta^2 \lambda_2)\mathbb{E}[\|X_k^i\|^2] + 2\eta R \mathbb{E}[\|X_k^i\|] + 2\eta(\eta(R^2 + \lambda_2 c_r) + \lambda d)$$
$$\leq (1 - \eta \lambda_1)\mathbb{E}[\|X_k^i\|^2] + 2\eta(\eta(R^2 + \lambda_2 c_r) + \lambda d + R^2/\lambda_1)$$
$$(\because 2R\mathbb{E}[\|X_k^i\|] \leq \lambda_1 \mathbb{E}[\|X_k^i\|^2]/2 + 2R^2/\lambda_1).$$
where we used the assumption $\eta \leq \lambda_1/(4\lambda_2)$. Then, by the Gronwall lemma, it holds that
$$\mathbb{E}[\|X_k^i\|^2] \leq (1 - \eta \lambda_1)^k \mathbb{E}[\|X_0^i\|^2] + \frac{1 - (1 - \eta \lambda_1)^k}{\eta \lambda_1} \eta[\eta(R^2 + \lambda_2 c_r) + \lambda d + R^2/\lambda_1]$$
$$\leq \mathbb{E}[\|X_0^i\|^2] + \frac{1}{\lambda_1}[\eta(R^2 + \lambda_2 c_r) + \lambda d + R^2/(\lambda_1)].$$
This with the assumption $\eta \leq \lambda_1/(4\lambda_2)$ yields the assertion. $\qquad \square$

**Lemma 2.** *Let* $\bar{R}^2 := \mathbb{E}[\|X_0^i\|^2] + \frac{1}{\lambda_1}\left[\left(\frac{\lambda_1}{4\lambda_2} + \frac{1}{\lambda_1}\right)(R^2 + \lambda_2 c_r) + \lambda d\right]$. *Under Assumptions 1 and 2, if $\eta \leq \lambda_1/(4\lambda_2)$ and we define*

$$\delta_\eta = 8[R^2 + \lambda_2(c_r + \bar{R}^2) + d]\bar{L}^2(\eta^2 + \lambda\eta),$$

*then it holds that*

$$\mathbb{E}_{\widetilde{\mathscr{X}_t},\widetilde{\mathscr{X}_0}}\left[\left\|\nabla\frac{\delta F(\tilde{\mu}_0^{(N)})}{\delta\mu}(\widetilde{X}_0^i) - \nabla\frac{\delta F(\tilde{\mu}_t^{(N)})}{\delta\mu}(\widetilde{X}_t^i)\right\|^2\right] \leq \delta_\eta.$$

*Proof.* The proof is basically a mean field generalization of Nitanda et al. (2022). By Assumptions 1 and 2, the first two terms in the right hand side can be bounded as

$$\left\|\nabla\frac{\delta F(\tilde{\mu}_0^{(N)})}{\delta\mu}(\widetilde{X}_0^i) - \nabla\frac{\delta F(\tilde{\mu}_t^{(N)})}{\delta\mu}(\widetilde{X}_t^i)\right\|$$

$$= \left\|\nabla\frac{\delta U(\tilde{\mu}_0^{(N)})}{\delta\mu}(\widetilde{X}_0^i) + \nabla r(\widetilde{X}_0^i) - \nabla\frac{\delta U(\tilde{\mu}_t^{(N)})}{\delta\mu}(\widetilde{X}_t^i) - \nabla r(\widetilde{X}_t^i)\right\|$$

$$\leq L(W_2(\tilde{\mu}_0^{(N)}, \tilde{\mu}_t^{(N)}) + \|\widetilde{X}_0^i - \widetilde{X}_t^i\|) + \lambda_2\|\widetilde{X}_t^i - \hat{X}_t^i\|. \tag{25}$$

Therefore, the right hand side can be further bounded as

$$\left\|\nabla\frac{\delta F(\tilde{\mu}_0^{(N)})}{\delta\mu}(\widetilde{X}_0^i) - \nabla\frac{\delta F(\tilde{\mu}_t^{(N)})}{\delta\mu}(\widetilde{X}_t^i)\right\|^2$$

$$\leq (L + \lambda_2)^2(W_2(\tilde{\mu}_0^{(N)}, \tilde{\mu}_t^{(N)}) + \|\widetilde{X}_0^i - \widetilde{X}_t^i\|)^2$$

$$\leq 2\bar{L}^2\frac{1}{N}\sum_{i=1}^N\|\widetilde{X}_0^i - \widetilde{X}_t^i\|^2 + 2\bar{L}^2\|\widetilde{X}_0^i - \widetilde{X}_t^i\|^2$$

$$\leq 2\bar{L}^2\frac{1}{N}\sum_{i=1}^N\left\|tv_k^i - \sqrt{2t\lambda}\xi_k^i\right\|^2 + 2\bar{L}^2\|tv_k^i - \sqrt{2t\lambda}\xi_k^i\|^2.$$

Then, by taking the expectation, it holds that

$$\mathbb{E}_{\widetilde{\mathscr{X}_t},\widetilde{\mathscr{X}_0}}\left[\left\|\nabla\frac{\delta F(\tilde{\mu}_0^{(N)})}{\delta\mu}(\widetilde{X}_0^i) - \nabla\frac{\delta F(\tilde{\mu}_t^{(N)})}{\delta\mu}(\widetilde{X}_t^i)\right\|^2\right]$$

$$\leq 2\bar{L}^2\frac{1}{N}\sum_{i=1}^N(t^2\mathbb{E}[\|v_k^i\|^2] + 2t\lambda d) + 2\bar{L}^2(t^2\mathbb{E}[\|v_k^i\|^2] + 2t\lambda d)$$

$$\leq 4\bar{L}^2[t^2 2(R^2 + \lambda_2(c_r + \bar{R}^2)) + 2t\lambda d] \quad (\because \text{Lemma } 1)$$

$$= 8\bar{L}^2[t^2(R^2 + \lambda_2(c_r + \bar{R}^2)) + t\lambda d]$$

$$\leq 8[R^2 + \lambda_2(c_r + \bar{R}^2) + d]\bar{L}^2(t^2 + t\lambda).$$

Then, by noticing $t \leq \eta$, we obtain the assertion. $\qquad\square$

## E.2 Wasserstein Distance Bound

Recall that $W_2(\mu,\nu)$ is the 2-Wasserstein distance between $\mu$ and $\nu$. We let $D(\mu,\nu) = \int\log\left(\frac{\nu}{\mu}\right)\mathrm{d}\nu$ be the KL-divergence between $\mu$ and $\nu$.

**Lemma 3.** *Under Assumptions 1 and 3, it holds that*

$$W_2^2(\mu^{(N)}, \mu^{*N}) \leq \frac{2}{\lambda\alpha}(\mathscr{F}^N(\mu^{(N)}) - N\mathscr{F}(\mu^*)).$$

*Proof.* By Assumption 3, $\mu^*$ satisfies the LSI condition with a constant $\alpha > 0$. Then, it is known that its tensor product $\mu^{*N}$ also satisfies the LSI condition with the same constant $\alpha$ (see, for example, Proposition 5.2.7 of Bakry et al. (2014)). Then, Otto-Villani theorem Otto and Villani (2000) yields the Talagrand's inequality of the tensorized measure $\mu^{*N}$:

$$W_2^2(\mu^{(N)}, \mu^{*N}) \leq \frac{2}{\alpha} D(\mu^{(N)}, \mu^{*N}).$$

Moreover, the proof of Theorem 2.11 of Chen et al. (2022) yields that, under Assumption 1, it holds that

$$D(\mu^{(N)}, \mu^{*N}) \leq \frac{1}{\lambda}(\mathscr{F}^N(\mu^{(N)}) - N\mathscr{F}(\mu^*)). \tag{26}$$

$\square$

**Lemma 4.** *Suppose that* $|h_x(z) - h_{x'}(z)| \leq L\|x - x'\|$ ($\forall x, x' \in \mathbb{R}^d$) *and let* $V_{\mu^*} := \mathrm{Var}_{\mu^*}(f_{\mu^*}) = \int (f_{\mu^*}(z) - h_x(z))^2 \mathrm{d}\mu^*(x)$, *then it holds that*

$$\mathbb{E}_{\mathscr{X}_k \sim \mu_k^{(N)}}[(f_{\mu_{\mathscr{X}_k}}(z) - f_{\mu^*}(z))^2] \leq \frac{2L^2}{N} W_2^2(\mu_k^{(N)}, \mu^{*N}) + \frac{2}{N} V_{\mu^*}.$$

*Proof.* Consider a coupling $\gamma$ of $\mu_{\mathscr{X}_k}^{(N)}$ and $\mu^{*N}$ and let $(\mathscr{X}_k, \mathscr{X}_*) = ((X_k^i)_{i=1}^N, (X_*^i)_{i=1}^N)$ be a random variable obeying the law $\gamma$. Then,

$$(f_{\mu_{\mathscr{X}_k}}(z) - f_{\mu^*}(z))^2 = \left( \frac{1}{N} \sum_{i=1}^N h_{X_k^i}(z) - \int h_x(z) \mathrm{d}\mu^*(x) \right)^2$$

$$= \left( \frac{1}{N} \sum_{i=1}^N (h_{X_k^i}(z) - h_{X_*^i}(z)) + \frac{1}{N} \sum_{i=1}^N h_{X_*^i}(z) - \int h_x(z) \mathrm{d}\mu^*(x) \right)^2$$

$$\leq 2 \left( \frac{1}{N} \sum_{i=1}^N (h_{X_k^i}(z) - h_{X_*^i}(z)) \right)^2 + 2 \left( \frac{1}{N} \sum_{i=1}^N h_{X_*^i}(z) - \int h_x(z) \mathrm{d}\mu^*(x) \right)^2$$

$$\leq 2 \left( \frac{1}{N} \sum_{i=1}^N L\|X_k^i - X_*^i\| \right)^2 + 2 \left( \frac{1}{N} \sum_{i=1}^N h_{X_*^i}(z) - \int h_x(z) \mathrm{d}\mu^*(x) \right)^2$$

$$\leq 2L^2 \frac{1}{N} \sum_{i=1}^N \|X_k^i - X_*^i\|^2 + 2 \left( \frac{1}{N} \sum_{i=1}^N h_{X_*^i}(z) - \int h_x(z) \mathrm{d}\mu^*(x) \right)^2.$$

Then, by taking the expectation of the both side with respect to $(\mathscr{X}_k, \mathscr{X}_*)$, we have that

$$\mathbb{E}_{\mathscr{X}_k \sim \mu_k^{(N)}}[(f_{\mu_{\mathscr{X}_k}}(z) - f_{\mu^*}(z))^2]$$

$$\leq 2L^2 \mathbb{E}_\gamma \left[ \frac{1}{N} \sum_{i=1}^N \|X_k^i - X_*^i\|^2 \right] + \frac{2}{N} V_{\mu^*}.$$

Hence, taking the infimum of the coupling $\gamma$ yields that

$$\mathbb{E}_{\mathscr{X}_k \sim \mu_k^{(N)}}[(f_{\mu_{\mathscr{X}_k}}(z) - f_{\mu^*}(z))^2] \leq \frac{2L^2}{N} W_2^2(\mu_k^{(N)}, \mu^{*N}) + \frac{2}{N} V_{\mu^*}.$$

$\square$

**Remark 2.** *By the self-consistent condition of* $\mu^*$, *we can see that* $\mu^*$ *is sub-Gaussian. Therefore,* $V_{\mu^*} < \infty$ *is always satisfied.*

### E.3 Logarithmic Sobolev Inequality

**Lemma 5.** *Under Assumptions 1 and 2, $\mu^*$ and $p_{\mathscr{X}}$ satisfy the LSI condition with a constant*

$$\alpha \geq \frac{\lambda_1}{2\lambda} \exp\left(-4\frac{R^2}{\lambda_1\lambda}\sqrt{2d/\pi}\right) \vee \left\{\frac{4\lambda}{\lambda_1} + \left(\frac{R}{\lambda_1} + \sqrt{\frac{2\lambda}{\lambda_1}}\right)^2 e^{\frac{R^2}{2\lambda_1\lambda}}\left[2 + d + \frac{d}{2}\log\left(\frac{\lambda_2}{\lambda_1}\right) + 4\frac{R^2}{\lambda_1\lambda}\right]\right\}^{-1}.$$

*Proof.* In the following, we give two lower bounds of $\alpha$. By taking the maximum of the two, we obtain the assertion.

(1) By Assumption 2, $\frac{\delta U(\mu)}{\delta\mu}$ is Lipschitz continuous with the Lipschitz constant $R$, which implies that $\frac{1}{\lambda}\frac{\delta U(\mu)}{\delta\mu}$ is $R/\lambda$-Lipschitz continuous. Hence, $\mu^*$ and $p_{\mathscr{X}}$ is a Lipschitz perturbation of $\nu(x) \propto \exp(-\lambda^{-1}r(x))$. Since $\lambda^{-1}\nabla\nabla^\top r(x) \succeq \frac{\lambda_1}{\lambda}I$ by Assumption 1, Miclo's trick (Lemma 2.1 of Bardet et al. (2018)) yields that $\mu^*$ and $p_{\mathscr{X}}$ satisfy the LSI with a constant

$$\alpha \geq \frac{\lambda_1}{2\lambda} \exp\left(-4\frac{\lambda}{\lambda_1}\left(\frac{R}{\lambda}\right)^2\sqrt{2d/\pi}\right) = \frac{\lambda_1}{2\lambda} \exp\left(-4\frac{R^2}{\lambda_1\lambda}\sqrt{2d/\pi}\right).$$

(2) We can easily check that $V(x) = \frac{r(x)}{\lambda}$ and $H(x) = \frac{1}{\lambda}\frac{\delta U(\tilde{\mu})}{\delta\mu}(x)$, for appropriately chosen $\tilde{\mu}$, satisfies the conditions in Lemma 6 with $c_1 = \frac{\lambda_1}{\lambda}$, $c_2 = \frac{\lambda_2}{\lambda}$, $c_V = c_r$, and $\bar{L} = \frac{R}{\lambda}$. Hence, $\mu^*$ and $p_{\mathscr{X}}$ satisfy the LSI with a constant $\alpha$ such that

$$\alpha \geq \left\{\frac{4\lambda}{\lambda_1} + \left(\frac{L}{\lambda_1} + \sqrt{\frac{2\lambda}{\lambda_1}}\right)^2 e^{\frac{R^2}{2\lambda_1\lambda}}\left[2 + d + \frac{d}{2}\log\left(\frac{\lambda_2}{\lambda_1}\right) + 4\frac{R^2}{\lambda_1\lambda}\right]\right\}^{-1}.$$

$\square$

**Lemma 6** (Log Sobolev inequality with Lipschitz perturbation). *Let $\nu(x) \propto \exp(-V(x))$ where $\nabla\nabla^\top V(x) \succeq c_1 I$, $x^\top \nabla V(x) \geq c_1\|x\|^2$ and $0 \leq V(x) \leq c_2(c_V + \|x\|^2)$ with $c_1, c_2, c_V > 0$, and let $H : \mathbb{R}^d \to \mathbb{R}$ is a Lipschitz continuous function with the Lipschitz constant $\bar{L}$. Suppose that $\mu \in \mathcal{P}$ is given by $\mu(x) \propto \exp(-H(x))\nu(x)$. Then, $\mu$ satisfies the LSI with a constanat $\alpha$ such that*

$$\alpha \geq \left\{\frac{4}{c_1} + \left(\frac{\bar{L}}{c_1} + \sqrt{\frac{2}{c_1}}\right)^2 e^{\frac{\bar{L}^2}{2c_1}}\left[2 + d + \frac{d}{2}\log\left(\frac{c_2}{c_1}\right) + 4\frac{\bar{L}^2}{c_1}\right]\right\}^{-1}.$$

*Proof.* Since $\nu$ is a strongly log-concave distribution, the Bakry–Emery argument (Bakry and Émery, 1985b) yields that it satisfies the LSI condition with a constant $\alpha' = c_1$ by the assumption $\nabla\nabla^\top V(x) \succeq c_1 I$.

Next, we evaluate the second moment of $\nu$ because it is required in the following analysis. Since we know that $\nu$ is the stationary distribution of the SDE $dX_t = -\nabla V(X_t)d + \sqrt{2}dW_t$, its corresponding infinitesimal generator gives that

$$\frac{d}{dt}\mathbb{E}[\|X_t\|^2] = \mathbb{E}[-2X_t^\top \nabla V(X_t)] + 2d \leq -2c_1\mathbb{E}[\|X_t\|^2] + 2d.$$

Then, by the Gronwall lemma, we have

$$\mathbb{E}[\|X_t\|^2] \leq \exp(-2c_1 t)\mathbb{E}[\|X_0\|^2] + \int_0^t \exp(-2c_1(t-s)))ds2d$$

$$= \exp(-2c_1 t)\mathbb{E}[\|X_0\|^2] + (1 - \exp(-2c_1 t))\frac{d}{c_1}.$$

Hence, by taking $X_0 = 0$ (a.s.), we see that $\limsup_t \mathbb{E}[\|X_t\|^2] \leq d/c_1$ that yields $\mathbb{E}_\nu[\|X^2\|] \leq d/c_1$.

A distribution $\tilde{\mu}$ satisfies the Poincaré inequality with a constant $\tilde{\alpha}$ if it holds that

$$\mathbb{E}_{\tilde{\mu}}[(f - \mathbb{E}_{\tilde{\mu}}[f])^2] \leq \frac{1}{\tilde{\alpha}}\mathbb{E}_{\tilde{\mu}}[\|\nabla f\|^2],$$

for all bounded $f$ with bounded derivatives. By Example (3) in Section 7.1 of Cattiaux and Guillin (2014), $\mu$ satisfies the Poincaré inequality with a constant $\tilde{\alpha}$ such that

$$\frac{1}{\tilde{\alpha}} \leq \frac{1}{2}\left(\frac{2\bar{L}}{c_1} + \sqrt{\frac{8}{c_1}}\right)^2 e^{\frac{\bar{L}^2}{2cc_1}}.$$

Here, let $G(x) = H(x) - H(0)$. We can see that $\mu(x)$ can be expressed as $\mu(x) = \frac{\exp(-G(x))\nu(x)}{Z_G}$ with a normalizing constant $Z_G > 0$. Note that, by the Lipschitz continuity assumption of $H$, we have

$$|G(x)| \leq \bar{L}\|x\|.$$

Hence, we have the following evaluation of the normalizing constant $Z_G$:

$$\begin{aligned}
Z_G &= \int \exp(-G(x))\nu(x)\mathrm{d}x \\
&\geq \int \exp\left(-\bar{L}\|x\| - c_2(c_V + \|x\|^2)\right)\frac{1}{\sqrt{(2\pi(1/c_1))^d}}\mathrm{d}x \\
&\geq \int \exp\left(-\frac{3c_2}{2}\|x\|^2 - \frac{\bar{L}^2}{2c_2} - c_2 c_V\right)\frac{1}{\sqrt{(2\pi(1/c_1))^d}}\mathrm{d}x \\
&= \left(\frac{c_1}{3c_2}\right)^{d/2}\exp\left(-\frac{\bar{L}^2}{2c_2} - c_2 c_V\right).
\end{aligned}$$

Theorem 2.7 of Cattiaux and Guillin (2022) claims that $\mu$ satisfies the LSI with a constant $\alpha$ such that

$$\frac{2}{\alpha} \leq \frac{(\beta+1)(1+\theta^{-1})}{\beta}\frac{2}{\alpha'} + \frac{1}{\tilde{\alpha}}(2 + \mathbb{E}_\nu[G - \log(Z_G)]) + \bar{L}^2\frac{2}{\tilde{\alpha}\alpha'}\left(\frac{(1+\theta)(1+\beta)}{4\beta} + \frac{\beta^2}{2}\right),$$

for any $\beta > 0$ and $\theta > 0$. The right hand side can be bounded by

$$\begin{aligned}
&\frac{(\beta+1)(1+\theta^{-1})}{\beta}\frac{2}{c_1} + \frac{1}{\tilde{\alpha}}\left(2 + \bar{L}\mathbb{E}_\nu[\|X\|] + \frac{\bar{L}^2}{2c_2} + \frac{d}{2}\log(3c_2/c_1)\right) + \bar{L}^2\frac{2}{\tilde{\alpha}c_1}\left(\frac{(1+\theta)(1+\beta)}{4\beta} + \frac{\beta^2}{2}\right) \\
\leq&\frac{(\beta+1)(1+\theta^{-1})}{\beta}\frac{2}{c_1} + \frac{1}{\tilde{\alpha}}\left(2 + \bar{L}\sqrt{\frac{d}{c_1}} + \frac{\bar{L}^2}{2c_1} + \frac{d}{2}\log(3c_2/c_1)\right) + \bar{L}^2\frac{2}{\tilde{\alpha}c_1}\left(\frac{(1+\theta)(1+\beta)}{4\beta} + \frac{\beta^2}{2}\right) \\
\leq&\frac{(\beta+1)(1+\theta^{-1})}{\beta}\frac{2}{c_1} + \frac{1}{\tilde{\alpha}}\left(2 + \frac{\bar{L}^2}{c_1} + \frac{d}{2}(\log(3c_2/c_1) + 1)\right) + \bar{L}^2\frac{2}{\tilde{\alpha}c_1}\left(\frac{(1+\theta)(1+\beta)}{4\beta} + \frac{\beta^2}{2}\right) \\
\leq&\frac{(\beta+1)(1+\theta^{-1})}{\beta}\frac{2}{c_1} + \frac{1}{\tilde{\alpha}}\left[2 + \frac{d}{2}(\log(c_2/c_1) + 2) + \frac{\bar{L}^2}{c_1}\left(1 + \frac{(1+\theta)(1+\beta)}{2\beta} + \beta^2\right)\right].
\end{aligned}$$

Hence, if we set $\beta = 1$ and $\theta = 1$, we have

$$\begin{aligned}
\frac{1}{\alpha} &\leq 2\frac{2}{c_1} + \frac{1}{2\tilde{\alpha}}\left[2 + \frac{d}{2}(\log(c_2/c_1) + 2) + \frac{\bar{L}^2}{c_1}(1 + 2 + 1)\right], \\
&\leq \frac{4}{c_1} + \left(\frac{\bar{L}}{c_1} + \sqrt{\frac{2}{c_1}}\right)^2 e^{\frac{\bar{L}^2}{2c_1}}\left[2 + d + \frac{d}{2}\log\left(\frac{c_2}{c_1}\right) + 4\frac{\bar{L}^2}{c_1}\right].
\end{aligned}$$

$\square$

### E.4 Uniform Log-Sobolev Inequality

**Lemma 7** (Uniform log-Sobolev inequality). *Under the same condition as Theorem 2, it holds that*

$$-\frac{\lambda^2}{2N}\sum_{i=1}^{N}\mathbb{E}_{\mathscr{X}\sim\tilde{\mu}_t^{(N)}}\left[\left\|\nabla_i\log\left(\frac{\tilde{\mu}_t^{(N)}}{p^{(N)}}(\mathscr{X})\right)\right\|^2\right] \leq -\frac{\lambda\alpha}{2}\left(\frac{1}{N}\mathscr{F}^N(\tilde{\mu}_t^{(N)}) - \mathscr{F}(\mu^*)\right) + \frac{C_\lambda}{N},$$

(27)

*with a constant $C_\lambda = 2\lambda L\alpha(1 + 2c_L\bar{R}^2) + 2\lambda^2 L^2\bar{R}^2$.*

*Proof.* The derivation is analogous to Chen et al. (2022), but we present the full proof for the sake of the completeness. For $\mathscr{X} = (X_i)_{i=1}^N$, let $\mathscr{X}^{-i} := (X_j)_{j \neq i}$. Then, it holds that

$$-\sum_{i=1}^N \mathbb{E}_{\mathscr{X} \sim \tilde{\mu}_t^{(N)}} \left[ \left\| \nabla_i \log \left( \frac{\tilde{\mu}_t^{(N)}}{p^{(N)}}(\mathscr{X}) \right) \right\|^2 \right]$$

$$= -\sum_{i=1}^N \mathbb{E}_{\mathscr{X} \sim \tilde{\mu}_t^{(N)}} \left[ \left\| \nabla_i \log \left( \tilde{\mu}_t^{(N)}(\mathscr{X}) \right) - \frac{1}{\lambda} \nabla \frac{\delta F(\mu_{\mathscr{X}})}{\delta \mu}(X_i) \right\|^2 \right]$$

$$= -\sum_{i=1}^N \mathbb{E}_{\mathscr{X} \sim \tilde{\mu}_t^{(N)}} \left[ \left\| \nabla_i \log \left( \tilde{\mu}_t^{(N)}(\mathscr{X}) \right) - \frac{1}{\lambda} \nabla \frac{\delta F(\mu_{\mathscr{X}^{-i}})}{\delta \mu}(X_i) + \frac{1}{\lambda} \nabla \frac{\delta F(\mu_{\mathscr{X}^{-i}})}{\delta \mu}(X_i) \right.\right.$$
$$\left.\left. - \frac{1}{\lambda} \nabla \frac{\delta F(\mu_{\mathscr{X}})}{\delta \mu}(X_i) \right\|^2 \right]$$

$$\leq -\frac{1}{2} \sum_{i=1}^N \mathbb{E}_{\mathscr{X} \sim \tilde{\mu}_t^{(N)}} \left[ \left\| \nabla_i \log \left( \tilde{\mu}_t^{(N)}(\mathscr{X}) \right) - \frac{1}{\lambda} \nabla \frac{\delta F(\mu_{\mathscr{X}^{-i}})}{\delta \mu}(X_i) \right\|^2 \right]$$

$$+ \sum_{i=1}^N \mathbb{E}_{\mathscr{X} \sim \tilde{\mu}_t^{(N)}} \left[ \left\| \frac{1}{\lambda} \nabla \frac{\delta F(\mu_{\mathscr{X}^{-i}})}{\delta \mu}(X_i) - \frac{1}{\lambda} \nabla \frac{\delta F(\mu_{\mathscr{X}})}{\delta \mu}(X_i) \right\|^2 \right]$$

$$\leq -\frac{1}{2} \sum_{i=1}^N \mathbb{E}_{\mathscr{X} \sim \tilde{\mu}_t^{(N)}} \left[ \left\| \nabla_i \log \left( \tilde{\mu}_t^{(N)}(\mathscr{X}) \right) - \frac{1}{\lambda} \nabla \frac{\delta F(\mu_{\mathscr{X}^{-i}})}{\delta \mu}(X_i) \right\|^2 \right]$$

$$+ \sum_{i=1}^N L^2 \mathbb{E}_{\mathscr{X} \sim \tilde{\mu}_t^{(N)}} [W_2^2(\mu_{\mathscr{X}}, \mu_{\mathscr{X}^{-i}})]$$

$$= -\frac{1}{2} \sum_{i=1}^N \mathbb{E}_{\mathscr{X} \sim \tilde{\mu}_t^{(N)}} \left[ \mathbb{E}_{\mathscr{X} \sim \tilde{\mu}_t^{(N)}} \left[ \left\| \nabla_i \log \left( \tilde{\mu}_t^{(N)}(\mathscr{X}) \right) - \frac{1}{\lambda} \nabla \frac{\delta F(\mu_{\mathscr{X}^{-i}})}{\delta \mu}(X_i) \right\|^2 \mid \mathscr{X}^{-i} \right] \right]$$

$$+ \sum_{i=1}^N L^2 \mathbb{E}_{\mathscr{X} \sim \tilde{\mu}_t^{(N)}} [W_2^2(\mu_{\mathscr{X}}, \mu_{\mathscr{X}^{-i}})],$$

where the second inequality is due to the Lipschitz continuity of $\nabla \frac{\delta U}{\delta \mu}$ in terms of the Wasserstein distance (Assumption 2). Here, the term corresponding to the Wasserstein distance can be upper bounded as

$$\mathbb{E}_{\mathscr{X} \sim \tilde{\mu}_t^{(N)}} [W_2^2(\mu_{\mathscr{X}}, \mu_{\mathscr{X}^{-i}})] \leq \mathbb{E}_{\mathscr{X} \sim \tilde{\mu}_t^{(N)}} \left[ \frac{1}{N(N-1)} \sum_{j \neq i} \| X_j - X_i \|^2 \right]$$

$$\leq \mathbb{E}_{\mathscr{X} \sim \tilde{\mu}_t^{(N)}} \left[ \frac{2}{N(N-1)} \sum_{j \neq i} \| X_j \|^2 + \frac{2}{N} \| X_i \|^2 \right]$$

$$\leq \frac{4}{N} \bar{R}^2.$$

Denote by $P_{X_i | \mathscr{X}^{-i}}$ the conditional law of $X_i$ conditioned by $\mathscr{X}^{-i}$ and denote by $P_{\mathscr{X}^{-i}}$ the marginal law of $\mathscr{X}^{-i}$ where the joint law of $\mathscr{X}$ is $\tilde{\mu}_t^{(N)}$. Then, it holds that

$$\nabla_i \log(\tilde{\mu}_t^{(N)}(\mathscr{X})) = \frac{\nabla_i (P_{X_i | \mathscr{X}^{-i}}(X_i) P_{\mathscr{X}^{-i}}(\mathscr{X}^{-i}))}{P_{X_i | \mathscr{X}^{-i}}(X_i) P_{\mathscr{X}^{-i}}(\mathscr{X}^{-i})} = \frac{\nabla_i P_{X_i | \mathscr{X}^{-i}}(X_i)}{P_{X_i | \mathscr{X}^{-i}}(X_i)} = \nabla_i \log(P_{X_i | \mathscr{X}^{-i}}(X_i)).$$

$$-\sum_{i=1}^N \mathbb{E}_{\mathscr{X}^{-i} \sim P_{\mathscr{X}^{-i}}} \left[ \mathbb{E}_{X_i \sim P_{X_i | \mathscr{X}^{-i}}} \left[ \left\| \nabla_i \log \left( P_{X_i | \mathscr{X}^{-i}}(X_i) \right) - \frac{1}{\lambda} \nabla \frac{\delta F(\mu_{\mathscr{X}^{-i}})}{\delta \mu}(X_i) \right\|^2 \mid \mathscr{X}^{-i} \right] \right]$$

$$\leq -2\alpha \sum_{i=1}^N \mathbb{E}_{\mathscr{X} \sim \tilde{\mu}_t^{(N)}} \left[ D(p_{\mathscr{X}^{-i}}, P_{X_i | \mathscr{X}^{-i}}) \right],$$

by the LSI condition of the proximal Gibbs measure (Assumption 3). Let $\nu_r$ be the distribution with the density $\nu_r \propto \exp(-r(x)/\lambda)$. Note that the proximal Gibbs measure is the minimizer of the linearized objective:

$$p_{\mathscr{X}^{-i}} = \underset{\mu \in \mathcal{P}}{\arg\min} \left\{ \int \frac{\delta U(\mu_{\mathscr{X}^{-i}})}{\delta \mu}(X_i)\mathrm{d}\mu + \mathbb{E}_\mu[r] + \lambda \mathrm{Ent}(\mu) \right\}$$

$$= \underset{\mu \in \mathcal{P}}{\arg\min} \left\{ \int \frac{\delta U(\mu_{\mathscr{X}^{-i}})}{\delta \mu}(X_i)\mathrm{d}\mu + \lambda D(\nu_r, \mu) \right\}.$$

Then, by the optimality of the proximal Gibbs measure, it holds that

$$\lambda D(p_{\mathscr{X}^{-i}}, P_{X_i|\mathscr{X}^{-i}})$$

$$= \int \frac{\delta U(\mu_{\mathscr{X}^{-i}})}{\delta \mu}(x_i)(P_{X_i|\mathscr{X}^{-i}} - p_{\mathscr{X}^{-i}})(\mathrm{d}x_i) + \lambda D(\nu_r, P_{X_i|\mathscr{X}^{-i}}) - \lambda D(\nu_r, p_{\mathscr{X}^{-i}})$$

$$\geq \int \frac{\delta U(\mu_{\mathscr{X}^{-i}})}{\delta \mu}(x_i)(P_{X_i|\mathscr{X}^{-i}} - \mu^*)(\mathrm{d}x_i) + \lambda D(\nu_r, P_{X_i|\mathscr{X}^{-i}}) - \lambda D(\nu_r, \mu^*). \qquad (28)$$

The expectation of the first term of the right hand side can be further evaluated as

$$\sum_{i=1}^N \mathbb{E}_{\mathscr{X} \sim \tilde{\mu}_t^{(N)}} \left[ \int \frac{\delta U(\mu_{\mathscr{X}^{-i}})}{\delta \mu}(x_i)(P_{X_i|\mathscr{X}^{-i}} - \mu^*)(\mathrm{d}x_i) \right]$$

$$= \sum_{i=1}^N \mathbb{E}_{\mathscr{X} \sim \tilde{\mu}_t^{(N)}} \left[ \int \frac{\delta U(\mu_{\mathscr{X}^{-i}})}{\delta \mu}(x_i)\delta_{X_i}(\mathrm{d}x_i) - \int \frac{\delta U(\mu_{\mathscr{X}^{-i}})}{\delta \mu}(x_i)\mu^*(\mathrm{d}x_i) \right]$$

$$\geq \sum_{i=1}^N \mathbb{E}_{\mathscr{X} \sim \tilde{\mu}_t^{(N)}} \left\{ \int \frac{\delta U(\mu_{\mathscr{X}})}{\delta \mu}(x_i)\delta_{X_i}(\mathrm{d}x_i) - \int \frac{\delta U(\mu_{\mathscr{X}})}{\delta \mu}(x_i)\mu^*(\mathrm{d}x_i) \right.$$

$$\left. - \frac{2L}{N} \left[ 2 + c_L(\|X_i\|^2 + \mathbb{E}_{X \sim \mu^*}[\|X\|^2] + \|X_i\|^2 + \frac{1}{N-1}\sum_{j \neq i}\|X_j\|^2) \right] \right\}$$

$$\geq N\mathbb{E}_{\mathscr{X} \sim \tilde{\mu}_t^{(N)}} \left[ \int \frac{\delta U(\mu_{\mathscr{X}})}{\delta \mu}(x)\mu_{\mathscr{X}}(\mathrm{d}x) - \int \frac{\delta U(\mu_{\mathscr{X}})}{\delta \mu}(x)\mu^*(\mathrm{d}x) \right] - 2L\left(2 + 4c_L\bar{R}^2\right)$$

$$\geq N\mathbb{E}_{\mathscr{X} \sim \tilde{\mu}_t^{(N)}} \left[ U(\mu_{\mathscr{X}}) - U(\mu^*) - 2L\left(2 + 4c_L\bar{R}^2\right) \right]$$

$$\geq N\mathbb{E}_{\mathscr{X} \sim \tilde{\mu}_t^{(N)}} \left[ U(\mu_{\mathscr{X}}) - U(\mu^*) \right] - 2L\left(2 + 4c_L\bar{R}^2\right)$$

$$\geq N\mathbb{E}_{\mathscr{X} \sim \tilde{\mu}_t^{(N)}} \left[ U(\mu_{\mathscr{X}}) - U(\mu^*) \right] - 4L\left(1 + 2c_L\bar{R}^2\right),$$

where the first inequality is due to Lemma 8 and the second inequality is due to Lemma 1 (we also notice that $\mathbb{E}_{X \sim \mu^*}[\|X\|^2] \leq \bar{R}^2$). The second term in the right hand side of Eq. (28) can be evaluated as

$$\sum_{i=1}^N \mathbb{E}_{\tilde{\mu}_t^{(N)}}[D(\nu_r, P_{X_i|\mathscr{X}^{-i}})] = \sum_{i=1}^N \mathbb{E}_{\tilde{\mu}_t^{(N)}}[\mathrm{Ent}(P_{X_i|\mathscr{X}^{-i}})] - \mathbb{E}_{\tilde{\mu}_t^{(N)}} \left[ \sum_{i=1}^N \log(\nu_r(X_i)) \right]$$

$$\geq \mathrm{Ent}(\tilde{\mu}_t^{(N)}) - \mathbb{E}_{\tilde{\mu}_t^{(N)}} \left[ \sum_{i=1}^N \log(\nu_r(X_i)) \right] = D(\nu_r^N, \tilde{\mu}_t^{(N)}),$$

where we used Lemma 3.6 of Chen et al. (2022) in the first inequality. Combining all of them, we arrive at

$$-\sum_{i=1}^N \mathbb{E}_{\mathscr{X} \sim \tilde{\mu}_t^{(N)}} \left[ \left\| \nabla_i \log\left( \frac{\tilde{\mu}_t^{(N)}}{p^{(N)}}(\mathscr{X}) \right) \right\|^2 \right]$$

$$\leq -\frac{\alpha}{\lambda} \left[ N\mathbb{E}_{\mathscr{X} \sim \tilde{\mu}_t^{(N)}}[U(\mu_{\mathscr{X}})] + \lambda D(\nu_r^N, \tilde{\mu}_t^{(N)}) - N(U(\mu^*) + \lambda D(\nu_r, \mu^*)) - 4L\left(1 + 2c_L\bar{R}^2\right) \right] + 4L^2\bar{R}^2$$

$$= -\frac{\alpha}{\lambda} \left[ N\mathbb{E}_{\mathscr{X} \sim \tilde{\mu}_t^{(N)}}[F(\mu_{\mathscr{X}})] + \lambda \mathrm{Ent}(\tilde{\mu}_t^{(N)}) - N(F(\mu^*) + \lambda \mathrm{Ent}(\mu^*)) - 4L\left(1 + 2c_L\bar{R}^2\right) \right] + 4L^2\bar{R}^2$$

$$= -\frac{\alpha}{\lambda} \left( \mathscr{F}^N(\tilde{\mu}_t^{(N)}) - N\mathscr{F}(\mu^*) \right) + \frac{4L\alpha}{\lambda}\left(1 + 2c_L\bar{R}^2\right) + 4L^2\bar{R}^2.$$

Then, we have the assertion with

$$C_\lambda = 2\lambda L\alpha(1 + 2c_L\bar{R}^2) + 2\lambda^2 L^2 \bar{R}^2.$$

$\square$

**Lemma 8.** *With the same setting as Lemma 7, it holds that*

$$\left| \frac{\delta U(\mu_{\mathscr{X}})}{\delta \mu}(x) - \frac{\delta U(\mu_{\mathscr{X}^{-i}})}{\delta \mu}(x) \right| \leq \frac{L}{N} \left[ 2 + c_L \left( 2\|x\|^2 + \|X_i\|^2 + \frac{1}{N-1} \sum_{j \neq i} \|X_j\|^2 \right) \right]$$

*Proof.* Let $\mu_{\mathscr{X},\theta} := \theta\mu_{\mathscr{X}} + (1-\theta)\mu_{\mathscr{X}^{-i}}$. Then, we see that

$$\left| \frac{\delta U(\mu_{\mathscr{X}})}{\delta \mu}(x) - \frac{\delta U(\mu_{\mathscr{X}^{-i}})}{\delta \mu}(x) \right| = \int_0^1 \left| -\frac{1}{N}\frac{\delta^2 U(\mu_{\mathscr{X},\theta})}{\delta\mu^2}(x, X_i) + \frac{1}{N(N-1)} \sum_{j \neq i} \frac{\delta^2 U(\mu_{\mathscr{X},\theta})}{\delta\mu^2}(x, X_j) \right| \mathrm{d}\theta$$

$$\leq \frac{L}{N} \left[ 2 + c_L \left( 2\|x\|^2 + \|X_i\|^2 + \frac{1}{N-1} \sum_{j \neq i} \|X_j\|^2 \right) \right],$$

where we used Assumption 2 (this is the only place where the boundedness of the second order variation of $U$ is used). Hence, we obtain the assertion. $\square$