# OpenReview forum: "Convergence of mean-field Langevin dynamics: time-space discretization, stochastic gradient, and variance reduction"
_NeurIPS.cc/2023/Conference — NeurIPS 2023 spotlight_

### Official Review · Reviewer_WF3Y · 2023-07-02

**Soundness:** 4 excellent
**Presentation:** 2 fair
**Contribution:** 3 good
**Rating:** 6
**Confidence:** 4

**Summary:**

This paper studies the convergence of various algorithms falling under the umbrella of *Mean-Field Langevin Dynamics* (MFLD). Those algorithms encompass diverse approximations of the usual Langevin dynamics
$$ dX_t = -\nabla\frac{\delta F(\mu_t)}{\delta \mu}(X_t) + \sqrt{2\lambda}dW_t $$
on three main points:
- when the measure $\mu_t$ is approximated by a finite-population version, i.e. the empirical measure $\mu_X$ of $(X_t^1, \dots X_t^N)$ all following the same dynamics,
- when the dynamics are discretized according to a given time-step $\eta$,
- and finally when we only have access to a noisy version $v_t^i$ of $\nabla\frac{\delta F(\mu_X)}{\delta \mu}(X^i)$.

The authors provide quantitative bounds for the convergence of $\mu_X$ to the minimum of an appropriately defined measure, which minimizes an entropy-regularized objective. They first provide a one-step bound in the general case, which is then turned into more precise convergence bounds depending on the version of $v_t^i$ considered.

**Strengths:**

This paper considers a very important and widely studied problem in the study of wide neural networks.The results obtained are impressive: even if the finite population approximation is borrowed from Chen et al., it is complemented by very precise and technical arguments about discretization and stochastic approximation, which makes this paper a major improvement on the state of the art.

The paper flows decently well, and even the appendix proof are nicely explained and made as easy to follow as possible (which is not a simple task, given their technicality). The significance of the paper is made clear by expanding on several well-studied instances of the general problem.

**Weaknesses:**

On the significance side, the main drawback of the paper is the necessity for two regularization terms (one strong convexity term in the objective function $U$, and the entropic regularization term), and the heavy dependency of the bounds on those two parameters. I am aware, unfortunately, that this is for now a common restriction for any time-independent bounds on Langevin dynamics.

The main drawback of this paper is that it is simply too technical for a Neurips submission. The presentation is good, but still suffers heavily from the 9-page (and even the 10-page, if accepted) limit, which forces a lot of inline math even for key equations (e.g. the log-Sobolev definition). This results in a very dense and sometimes hard to follow article, which could really benefit from a longer exposition. Notably, some examples (especially the first one) could be expanded upon, both to understand exactly the role of first-variation functional and the comparison with existing results on SGD/mean-field for 2LNNs.

All in all, this paper is more suited for a (very good) journal than for Neurips itself; however, I cannot recommend rejection based on the significance of the results.

Some parts of the appendix are also a bit rushed, especially Appendix A, which only lists the necessary conditions for the examples to fit Assumptions 1 and 2 without any proof.

**Questions:**

- When you remove the entropic regularization term, the existence of a limit measure for the dynamics is not guaranteed anymore; however, some papers (e.g. Chizat and Bach '18) manage to recover results by assuming that there is still a convergence. Do you think this could also be the case for your discretization results ?
- Can your methods be applied to directly show an approximation result similar to (Mei et al. '18), i.e. an approximation bound between $\mu_t$ and $\mu_{X_t}$ instead of a direct control on $\mathcal F(\mu)$ ?

---

> ### Author Rebuttal · Authors · 2023-08-09
>
> Thank you very much for your insightful comments. We address the technical points below.
>
> **Q:** *Some parts of the appendix are also a bit rushed, especially Appendix A, which only lists the necessary conditions for the examples to fit Assumptions 1 and 2 without any proof.*
> **A:** Thank you very much for pointing it out.
> We will extend some parts of Appendix including justifications of Assumptions 1 and 2 in the revision.
>
> **Q:** *some papers (e.g. Chizat and Bach '18) manage to recover results by assuming that there is still a convergence. Do you think this could also be the case for your discretization results?*
> **A:** Thank you for raising an interesting question. Indeed, Chizat and Bach (2018) showed global convergence by assuming convergence in the first place. That is, convergence is guaranteed only when the activation function is homogeneous and the solution satisfies some specific conditions which cannot be ensured beforehand (more precisely, if the solution converges to some distribution in $W_2$ distance, then the limit distribution is the global optimal, however whether the solution converges or not is not guaranteed).
> In contrast, in our setting global convergence is guaranteed due to the entropy regularization, similar to (Mei et al. 2018). If we completely remove the regularization, then the quantitative convergence rate (which is exponential) would be lost unless additional assumptions are imposed, and consequently, the uniform-in-time propagation of chaos estimate would be very challenging to establish. We intend to investigate this problem in the future.
>
>
> **Q:** *Can your methods be applied to directly show an approximation result similar to (Mei et al. '18), i.e. an approximation bound between $\mu_t$ and $\mu_{X_t}$ instead of a direct control on $\mathcal{F}(\mu)$?*
> **A:** Yes, we have a convergence of $\mu_{X_t}$ and $\mu_t$ to $\mu^*$ in terms of the Wasserstein distance, which also yields convergence of the Wasserstein distance between $\mu_{X_t}$ and $\mu_t$ by the triangular inequality. Once again, the main difference from Mei et al. (2018) is that their discretization error blows up exponentially over the time, whereas our bound remains stable for all $t>0$.
>
> We would be happy to clarify any concerns or answer any questions that may come up during the discussion period.

---

> > ### Comment · Reviewer_WF3Y · 2023-08-15
> >
> > Thank you for your instructive answers. I still think this would be more suited for a journal publication, hence I will maintain my grade, but this is a very good set of results.

---

### Official Review · Reviewer_Y8on · 2023-07-07

**Soundness:** 3 good
**Presentation:** 3 good
**Contribution:** 3 good
**Rating:** 6
**Confidence:** 4

**Summary:**

This work considers the analysis of mean field langevin dynamics as implemented algorithmically. i.e,
a) particle approximation b) time discretization and c) stochastic gradients. Under the assumption of certain logarithmic sobolev inequalities, prior works were mostly restrict

**Strengths:**

Extensive and explicit non-asymptotic convergence rates are obtained for the practically implemented algorithm with finite particles, time discretization and stochastic gradients, which is a great addition to the literature.

**Weaknesses:**

1. What are the main technical insights in this work? It seems like an extension of Wibisono and Vempala's LSI analysis of LMC while accounting for the mean-field and stochastic gradients.

2. Assumption 4 seems non-standard and the existence of a.s. bounded large order derivatives of the stochastic gradient seems restrictive compared to the assumptions in SGD/ SGLD literature. Similarly Assumption 5 seems too restrictive.

3. What is the point of SVRG type algorithm when straightforward stochastic approximation like SGLD itself outperforms this? This is true in Table 1 for MFLD. We also refer to the Theorem 6 in [A1] for results on regular Langevin dynamics.

4. Can you compare your technique with the recent results which utilized the CLT structure in the batched noise to obtain sharp analysis of stochastic approximations such as SGLD? (see [A1]). In Theorem 3 in the current work, with a batchsize $B$, the error bounds will have a $\frac{\eta}{B}$ (similar to [A2]) Whereas [A1] gets a rate of $\frac{\eta}{B^2}$.  Since the analysis techniques are close to that of SGLD, and one of the main contributions is the addition of stochastic gradients, it would be good if the authors compare the results to that of [A0,A1,A2] on a technical level.

[A0] Non-convex learning via Stochastic Gradient Langevin Dynamics: a nonasymptotic analysis

[A1] Utilising the CLT Structure in Stochastic Gradient based Sampling: Improved Analysis and Faster Algorithms

[A2] Faster Convergence of Stochastic Gradient Langevin Dynamics for Non-Log-Concave Sampling


Minor: Theorem 2, definition of $\mathcal{Y}_k$ needs to have $\eta_k$
The notation is a bit cluttered at many points, which makes the paper hard to read sometimes.


I will give a borderline accept for now. Happy to improve my score once the authors provide a satisfactory response.

**Questions:**

See the weaknesses section

**Limitations:**

Yes.

---

> ### Author Rebuttal · Authors · 2023-08-09
>
> Thank you very much for your insightful comments.
>
> **Q:** *What are the main technical insights in this work? It seems like an extension of Wibisono and Vempala's LSI analysis of LMC while accounting for the mean-field and stochastic gradients.*
> **A:** First, we note that the analysis of the mean-field extension of Langevin dynamics is far from trivial, even in the continuous-time and infinite-width limit. As for the discretized algorithm, we need to establish a uniform-in-time propagation of chaos, which is technically challenging as we have to handle (1) the finite particle approximation, (2) the stochastic gradient approximation, and (3) the time discretization. Note that such propagation of chaos calculation is not required in Wibisono and Vempala's analysis due to the absence of mean-field interaction.
> Our main contribution is to derive these errors in one unified framework. In particular, (1) and (2) are technically demanding, and we obtained faster rate than existing work taking into account these approximation errors with an additional smoothness condition.
> Please refer to our Introduction section for a more detailed summary of our contribution.
>
> **Q:** *Assumption 4 seems non-standard and the existence of a.s. bounded large order derivatives of the stochastic gradient seems restrictive compared to the assumptions in SGD/SGLD literature.  Similarly Assumption 5 seems too restrictive.*
> **A:** Please note that our theorem also holds without Assumption 4. What we intend to convey is that the convergence rate can be further improved if we assume Assumption 4 additionally.
> Indeed, if we looked at the statement of Theorem 2, there are two types of bounds: one is without Assumption 4 and the other is with Assumption 4. We will make this point more explicit in the revision.
> Also, Assumptions 5 and 6 are not new and isolated assumptions, but rather, sufficient conditions for Assumption 4 when applied to different gradient estimators; and similarly, we have a valid convergence rate without these assumptions. Hence, our analysis includes the standard SGLD setting as a special case.
>
> **Q:** *Can you compare your technique with [A1]?*
> **A:** Thank you very much for letting us know a relevant paper.
> Indeed, this paper is quite interesting and should be taken into account.
> Our bound (Theorem 2) is so general that the stochasticity is induced by not only the minibatch selection but also any other sources of randomness. Hence, we cannot directly compare with their result.
> On the other hand, for certain special settings (SGD-MFLD and SVRG-MFLD), we can adapt their result into our analysis.
> That is, we may replace the term $\sigma_{v,k}^2$ by their evaluation of the square of the conditional expectation of the stochastic gradient, which would give better bound than our naive bound. On the other hand, if we assume Assumption 4, then for the SVRG setting,
> the dependency on $\epsilon$ is $\min\\{\sqrt{n/\epsilon},1/\epsilon\\}$ while their bound only achieves $1/\epsilon$. Hence, we have better dependency on $\epsilon$ in a small $\epsilon$ regime with a stronger assumption.
>
> That being said, we have the following concern on the proof reasoning in [A1].
> In the proof of Lemma 2 which is the key lemma to obtain a tighter bound, it is argued that $N_1,\dots,N_B$
> are independent and identical even after conditioning on $x_{k\eta}$ and $x_t$.
> However, it seems that this is not true because permutation invariance does not necessarily result in independence.
> More precisely, it should be mentioned that the marginal distribution of $N_i$ are the same and the expectation of their average can be replaced by expectation of one of them. (We think the assertion itself is correct.)
>
> We would be happy to clarify any concerns or answer any questions that may come up during the discussion period.

---

> > ### Comment · Reviewer_Y8on · 2023-08-19
> > **Thanks for the Response**
> >
> > Thank you for the response. This clarifies my questions. Bumping my score to a 6.

---

### Official Review · Reviewer_rhAi · 2023-07-07

**Soundness:** 4 excellent
**Presentation:** 4 excellent
**Contribution:** 4 excellent
**Rating:** 8
**Confidence:** 4

**Summary:**

In the work authors study mean field Langevin dynamics under stochastic gradient updates and prove uniform in time propagation of chaos that takes into account discretisation, stochastic errors which allows to establish convergence rates of MFLD.

**Strengths:**

1. Strong theory with explicit bounds
2. Excellent outline of results
3. Thorough comparison of convergence rates with other approaches
4. Wide applicability of results to different sg estimators and neural networks in mean field regime.

**Weaknesses:**

Seeing some numerical evaluation could make results even stronger to support the theory.

**Questions:**

-

**Limitations:**

Yes

---

> ### Author Rebuttal · Authors · 2023-08-09
>
> Thank you for the positive evaluation.
> Please find our answer to your concerns.
>
> **Q:** Seeing some numerical evaluation could make results even stronger to support the theory.
> **A:** Thank you very much for your suggestion.
> Since the theoretical part already occupies the whole part of the paper, we chose not to include the numerical experiments in the main text. Following your suggestion, we will add numerical experiments in the appendix. Indeed, our preliminary experiments shows good convergence with respect to the number of particles and the stochastic gradient also converges properly. We would like to add more thorough experiments in the final version.
>
> We would be happy to clarify any concerns or answer any questions that may come up during the discussion period.

---

> > ### Comment · Area_Chair_hNxr · 2023-08-20
> >
> > Dear reviewer,
> >
> > The author-reviewer discussion period ends in 2 days. Please review the authors' rebuttal and engage with them if you have additional questions or feedback. Your input during the discussion period is valued and helps improve the paper.
> >
> > Thanks, Area Chair

---

> > > ### Comment · Reviewer_rhAi · 2023-08-20
> > >
> > > Thank you authors for the reply.
> > >
> > > After reading other reviews and looking into the paper in more detail I've raised my score as this is indeed a very interesting paper that is worth publication.

---

### Official Review · Reviewer_wZfS · 2023-07-16

**Soundness:** 3 good
**Presentation:** 3 good
**Contribution:** 3 good
**Rating:** 7
**Confidence:** 4

**Summary:**

This paper studies the mean-field Langevin dynamics (MFLD) with stochastic gradient updates. In particular, the authors propose a general framework to prove a uniform-in-time propagation of chaos for MFLD. The authors establish the convergence rate guarantees to the regularized global optimal solution, simultaneously addressing the uses of particle approximation, time discretization and stochastic gradient.


**Strengths:**

This work provides a quite general framework for analyzing the mean-field Langevin dynamics (MFLD) with stochastic gradient updates. The authors establish the convergence rate guarantees to the regularized global optimal solution, simultaneously addressing the uses of particle approximation, time discretization and stochastic gradient. The results are interesting and appear to be novel. Various examples and practical implementations of MFLD are also provided to demonstrate the effectiveness of the proposed framework.

**Weaknesses:**

Purely theoretical paper. No numerical experiments are provided.


**Questions:**

It is suggested that the authors should provide some simple numerical experiments to demonstrate the effectiveness of the proposed algorithms.

**Limitations:**

Yes.

---

> ### Author Rebuttal · Authors · 2023-08-09
>
> Thank you for your positive evaluation. We address the technical comments below.
>
> ### Weaknesses:
> **Q:** Purely theoretical paper. No numerical experiments are provided.
> **A:** Since the theoretical part already occupies the whole part of the paper, we chose not to include the numerical experiments in the main text. Following your suggestion, we will add numerical experiments in the appendix. Indeed, our preliminary experiments showed good convergence with respect to the number of particles even with stochastic gradient approximation.
>
> ### Questions:
> **Q:** It is suggested that the authors should provide some simple numerical experiments to demonstrate the effectiveness of the proposed algorithms.
> **A:** Thank you for your suggestion. We have already done a preliminary experiment.
> We would like to include more thorough experiments in the appendix of the camera-ready version.
>
> We would be happy to clarify any concerns or answer any questions that may come up during the discussion period.

---

> > ### Comment · Area_Chair_hNxr · 2023-08-20
> >
> > Dear reviewer,
> >
> > The author-reviewer discussion period ends in 2 days. Please review the authors' rebuttal and engage with them if you have additional questions or feedback. Your input during the discussion period is valued and helps improve the paper.
> >
> > Thanks, Area Chair

---

> > ### Comment · Reviewer_wZfS · 2023-08-20
> > **Reply to Rebuttal**
> >
> > This is overall a very interesting work, regardless of the addition of numerical experiments. Adding experiments is just a suggestion. Also as a more theoretical researcher, I personally think that the theoretical contribution of the paper is sufficient for me to champion its acceptance. With numerical experiments, I would say the goal is to reach a broader audience and raise the impact of the paper, especially for large conferences like NeurIPS.

---

### Official Review · Reviewer_YwhW · 2023-07-27

**Soundness:** 4 excellent
**Presentation:** 4 excellent
**Contribution:** 4 excellent
**Rating:** 8
**Confidence:** 4

**Summary:**

This paper provides a set of results to analyze convergence of mean-field Langevin dynamics with a set of related algorithms. These results are in discrete time and space. Using log-Sobolev inequality techniques from optimal transport, which creates the opportunity for extensions to other settings. The propagation of chaos results are also proved in discrete time.

**Strengths:**

My feeling is that this paper provides a useful advance in its effort to systematically control discretization error in the propagation of chaos. I am not familiar with other results that have done this. The rather explicit results obtained for the mean-field dynamics settings illustrates favorable scaling for single loop algorithms, at least in sufficiently smooth problems.

**Weaknesses:**

I think the differences from Nitanda and Chizat could be articulated more clearly.

**Questions:**

Assumption 4 is rather complicated. Can the motivation for it be explained more clearly in the text?



**Limitations:**

LSI conditions are obviously hard to satisfy for some problems, but this is well-known and well-discussed in the paper.

---

> ### Author Rebuttal · Authors · 2023-08-09
>
> Thank you for your supportive comments. We address the technical comments below.
>
> ### Weaknesses
> **Q:** I think the differences from Nitanda and Chizat could be articulated more clearly.
> **A:** Thanks for your suggestion. Please notice that the differences from Nitanda et al. (2022) and Chizat (2022) are summarized in the Introduction section. That is, they did not derive (1) the finite particle approximation error of MFLD and (2) the stochastic gradient approximation error. On the other hand, we established a unifying framework to evaluate the time discretization error, the finite particle approximation, and the stochastic gradient approximation. One of our contributions is to derive a tight bound on the stochastic gradient approximation error in the propagation of chaos analysis, which is challenging because it requires the evaluation of correlation between the randomness of each gradient and the updated distribution.
>
> ### Questions
> **Q:** Assumption 4 is rather complicated. Can the motivation for it be explained more clearly in the text?
> **A:** Assumption 4 is essentially a higher-order smoothness condition, which is required to obtain an improved bound on the stochastic gradient approximation, i.e., smoother loss yields better stochastic approximation -- we will discuss this in more details in the revision.
> As an example, Assumption 5 (ii) and Assumption 6 (ii) are sufficient conditions for Assumption 4 in each corresponding situation (gradient estimator). We believe that these conditions are rather intuitive because they only impose boundedness of the second order derivatives.
>
> We would be happy to clarify any concerns or answer any questions that may come up during the discussion period.

---

> > ### Comment · Area_Chair_hNxr · 2023-08-20
> >
> > Dear reviewer,
> >
> > The author-reviewer discussion period ends in 2 days. Please review the authors' rebuttal and engage with them if you have additional questions or feedback. Your input during the discussion period is valued and helps improve the paper.
> >
> > Thanks, Area Chair

---

### Decision · Program_Chairs · 2023-09-21

**Decision:**

Accept (spotlight)

**Comment:**

This paper show non-asymptotic convergence of the mean-field Langevin dynamics (MFLD), handling discretization errors due to particle approximation, time discretization, and stochastic gradients. This is an important problem, especially given the recent interest in MFLD. The analysis is applicable to general settings, and I agree with the reviewers that that this paper is an important contribution to the literature.